# Improvement of a synthetic live bacterial therapeutic for phenylketonuria with biosensor-enabled enzyme engineering

Kristin J. Adolfsen[1], Isolde Callihan[1], Catherine E. Monahan [2], Per Jr. Greisen[1,3], James Spoonamore[1], Munira Momin[2], Lauren E. Fitch [1], Mary Joan Castillo[2], Lindong Weng[1,4], Lauren Renaud[2], Carl J. Weile[1], Jay H. Konieczka[1], Teodelinda Mirabella[2], Andres Abin-Fuentes[2], Adam G. Lawrence[1] & Vincent M. Isabella [2✉]

In phenylketonuria (PKU) patients, a genetic defect in the enzyme phenylalanine hydroxylase (PAH) leads to elevated systemic phenylalanine (Phe), which can result in severe neurological impairment. As a treatment for PKU, *Escherichia coli* Nissle (EcN) strain SYNB1618 was developed under Synlogic's Synthetic Biotic™ platform to degrade Phe from within the gastrointestinal (GI) tract. This clinical-stage engineered strain expresses the Phe-metabolizing enzyme phenylalanine ammonia lyase (PAL), catalyzing the deamination of Phe to the non-toxic product *trans*-cinnamate (TCA). In the present work, we generate a more potent EcN-based PKU strain through optimization of whole cell PAL activity, using biosensor-based high-throughput screening of mutant PAL libraries. A lead enzyme candidate from this screen is used in the construction of SYNB1934, a chromosomally integrated strain containing the additional Phe-metabolizing and biosafety features found in SYNB1618. Head-to-head, SYNB1934 demonstrates an approximate two-fold increase in in vivo PAL activity compared to SYNB1618.

[1] Zymergen Inc. (formerly enEvolv Inc.), 100 Acorn Park Drive, Cambridge, MA 02140, USA. [2] Synlogic Inc, 301 Binney St, Cambridge, MA 02139, USA. [3] Present address: Novo Nordisk Research Center Seattle Inc, 530 Fairview Ave N, Seattle, WA 98109, USA. [4] Present address: Sana Biotechnology, 1 Tower Place Suite 500, South San Francisco, CA 94080, USA. ✉email: vincent@synlogictx.com

The metabolic disease phenylketonuria (PKU) is characterized by the absence or disrupted function of the phenylalanine hydroxylase (PAH) enzyme. Patients with PKU are unable to metabolize the amino acid phenylalanine (Phe), resulting in its systemic accumulation. Elevated levels of Phe can lead to severe neurological complications and disability if left untreated. Fortunately, PKU has been diagnosed through newborn screening for over 50 years, and if the disease is detected and treated within the first weeks of life, severe disabilities can be mitigated[1].

As Phe is an essential amino acid, its intake in patients can be controlled through dietary restriction of protein; however, many patients fail to achieve lasting compliance due to the diet's stringent nature. Indeed, the discontinuation of the PKU diet into adulthood has been linked to a marked reduction in executive function and major depressive disorder[2,3]. In addition to controlling Phe intake through dietary management, two approved pharmacologic treatment options exist. One of these options, sapropterin dihydrochloride (KUVAN, BioMarin Pharmaceutical), is a PAH activator for patients with tetrathydrobiopterin (BH$_4$)-responsive PKU. Sapropterin is a BH$_4$ analog that provides additional cofactor and/or promotes productive folding of mutant PAH enzyme. However, due to its mechanism, sapropterin is only efficacious in the subset of PKU patients that have residual PAH activity, which is estimated to be approximately one-third of the PKU population[4,5]. Even in BH$_4$-responsive patients, sapropterin is rarely deployed as sole therapy and is meant to be combined with parallel dietary management[6]. The other treatment option, Pegvaliase (Palynziq, BioMarin Pharmaceutical), consists of a systemically injected pegylated Phe-degrading enzyme, phenylalanine ammonia lyase (PAL). However, this therapy has been associated with reports of severe immune-mediated adverse reactions and anaphylaxis[7,8]. There remains a need for safe orally delivered therapeutic treatment options for PKU patients that will address the needs of the entire patient population regardless of age or genetic background.

There has been significant recent interest in the application of synthetic biology to the design and engineering of probiotic organisms that can exert therapeutic functions from within the gastrointestinal (GI) tract[9]. We have previously reported the construction and characterization of the live biotherapeutic product (LBP) SYNB1618 for the treatment of PKU[10]. This genetically modified strain of *Escherichia coli* Nissle 1917 (EcN) expresses enzymes, including PAL (StlA) from *Photorhabdus luminescens* and L-amino acid deaminase (LAAD) from *Proteus mirabilis*[11], intended to degrade dietary and enterrorecirculating Phe into non-toxic metabolites within the GI tract (cinnamic acid[12,13] and phenylpyruvate[14], respectively). SYNB1618 was reported to significantly lower plasma Phe in Pah$^{enu2/enu2}$ mice (a mouse model of PKU) as well as to significantly prevent a spike in plasma Phe in healthy non-human primates (NHPs) administered a protein bolus[10]. SYNB1618 was evaluated in a randomized, placebo-controlled first-in-human study in healthy volunteers and PKU patients, where the strain was found to be safe and well-tolerated, with dose-responsive increases in SYNB1618-specific Phe metabolites in plasma and urine[15].

As enthusiasm for engineered LBPs advances, so too will the interest in methodologies employed to enhance their performance. For organisms such as *E. coli*, a multitude of modular parts exist for the construction of transcriptional gene circuits that allow for the tunable expression of engineered effector function[9]. Enhancing heterologous gene expression in engineered strains is a straightforward approach to optimize activity and is the first potential bottleneck addressed when seeking to maximize strain function. Enhancing strain function beyond gene expression relies on the application of other existing or evolving technologies.

Biosensors have emerged as a promising approach to relieve throughput limitation in metabolic and protein engineering[16]. When controlling a reporter signal such as growth or fluorescence, biosensors allow for the rapid enrichment of library sizes not previously accessible. Screening rates can reach 10$^7$ per hour when fluorescence is used as the reporter and cells are phenotyped on a fluorescence-activated cell sorter (FACS)[17]. To date, biosensor screening efforts have focused on deploying allosteric transcription factors (aTFs) found in nature to improve the production of metabolites as diverse as the flavonoid antioxidant naringenin to the rare non-toxic sugar psicose[18–23]. Although aTFs have been successfully engineered to respond to novel metabolites[24–26], to our knowledge there are no published accounts describing the use of engineered aTFs to improve commercial or therapeutic strains.

In this work, we first engineered an aTF to sense and respond to the presence of trans-cinnamate (TCA), the product of Phe catabolism by PAL. Using this biosensor as a means of both screening and selection, we describe the directed evolution of StlA, the PAL in our LBP, for enhanced whole-cell activity. Importantly, screening and selection could be performed in EcN so that the enhancement of PAL activity was assessed in the context of a therapeutically relevant strain. A candidate PAL variant was used to construct a second-generation PKU therapeutic strain, SYNB1934, which was shown to outperform SYNB1618 in vitro and in vivo.

## Results

**Whole-cell activity is not improved by increased expression of PAL**. PAL is a relatively uncomplicated enzyme that has no dependence on oxygen, redox equivalents, or cofactors[27]. However, in a whole-cell context, PAL activity may be influenced by a multitude of physiological or environmental factors, including bacterial metabolic activity (to drive Phe in or TCA out of cells), external pH, or end-product inhibition. Preliminary work sought to determine if strain activity was rate-limited by PAL expression, as this would offer a simple route for strain optimization. However, TCA production plateaued with increasing PAL expression (Supplementary Fig. 1). We, therefore, focused efforts towards optimizing the performance rather than the expression of the PAL enzyme itself in whole-cell format.

**TCA biosensor development and demonstration for whole-cell PAL activity screening**. To enable the screening of high-complexity PAL libraries for improved Phe degradation (TCA production), a specific TCA-responsive biosensor was engineered using Zymergen's proprietary sensor engineering platform. In the absence of TCA, the engineered aTF represses expression of a gene (labeled *gfp* in Fig. 1a) that encodes for the fluorescent reporter protein (green fluorescent protein, GFP); when TCA is introduced to the system, the repression is relieved, and *gfp* is expressed (Fig. 1a). This sensor system was transformed as a high-copy plasmid into a wild-type EcN background, and demonstrated a rank-ordered GFP signal readout to TCA produced by a ladder of PAL variants from early engineering efforts, expressed from low-copy constructs (Fig. 1b). For demonstration purposes, each of the strains in Fig. 1b was cultured separately in test tubes and analyzed for per cell GFP signal on a flow cytometer upon saturation. This strain ladder approach does not take into account any strain crosstalk that might be observed in a pooled library (e.g., low TCA producers responding to TCA in the bulk media produced by improved production phenotypes). Significant enrichment of a high-complexity pooled library, which

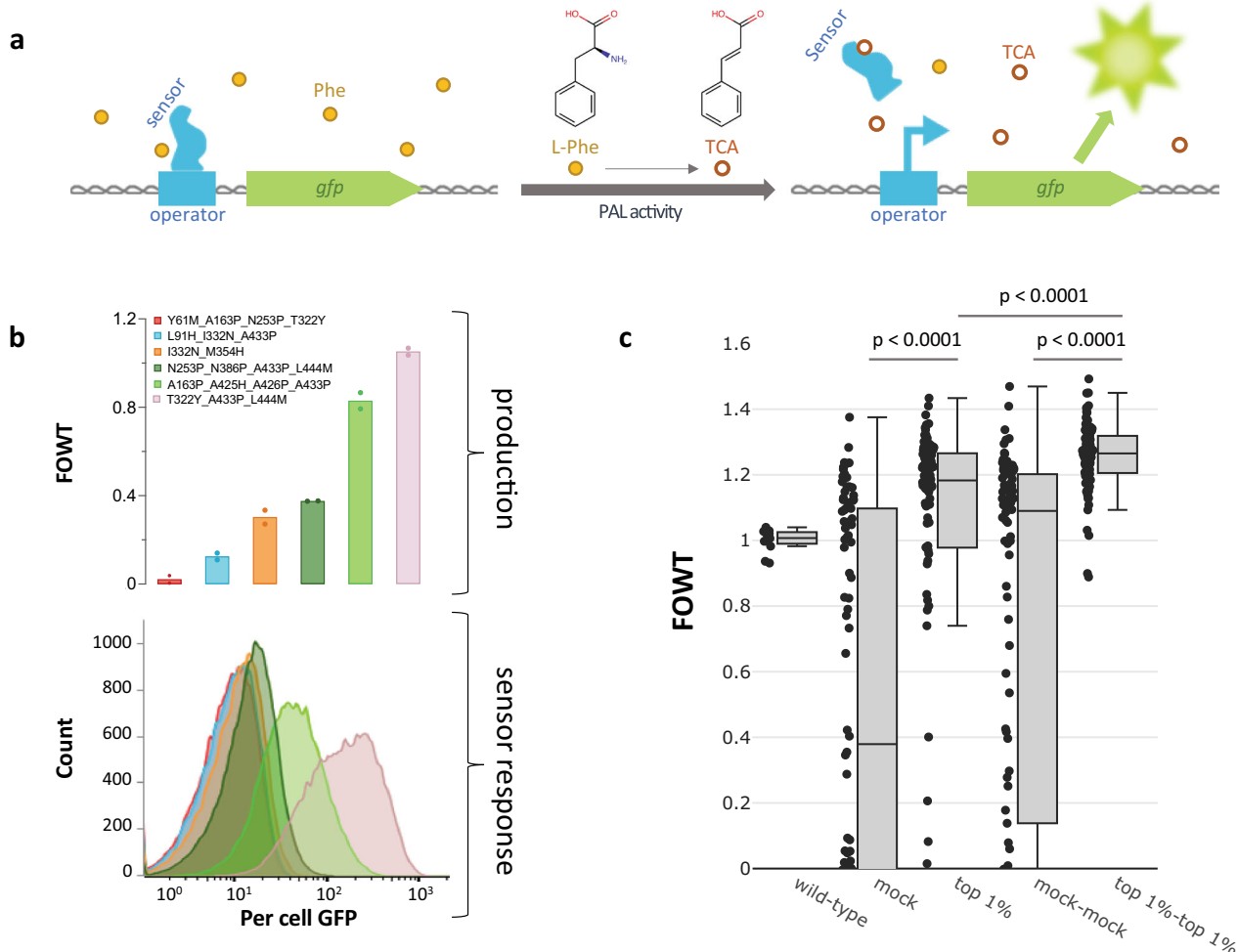

**Fig. 1 TCA biosensor description and demonstration. a** Schematic representation of biosensor-regulated reporter expression. In the absence of TCA, the allosteric transcription factor (aTF, labeled sensor in figure) binds the DNA operator site and prevents expression of the reporter gene *gfp* that encodes for a green fluorescent protein (GFP). PAL activity converts ʟ-Phe into TCA; TCA binds to the sensor aTF, causing a conformational change to relieve DNA binding (repression) and allowing the expression of the reporter gene. Symbols and abbreviations: phenylalanine (ʟ-Phe, yellow filled circles), *trans*-cinnamic acid (TCA, orange empty circles), PAL (phenylalanine ammonia lyase). **b** Top: whole-cell activity (see Methods, plate-based) of six StlA PAL variants, normalized to wild-type StlA activity (fold-over wild-type, FOWT, wild-type values can be found in the Source Data). $n = 2$ biological replicates, individual data points shown. Bottom: per cell GFP histograms of variants from the top panel following growth to saturation with simultaneous *stlA* (gene encoding for StlA) variant expression, sensor response, and *gfp* (gene encoding for GFP) expression. Colors are consistent between graphs. $n = 1$ representative culture per strain with 50,000 cells analyzed. **c** Enrichment demonstration of designed PAL library. Boxplots showing FOWT activity from different sort populations, with accompanying clonal data points. Boxes extend from the first to the third quartile, with a centerline indicating the median. Upper and lower whiskers extend to the maximum and minimum point, respectively, within 1.5 times the interquartile range of the bounds of the box. Data points outside of the whiskers are considered outliers. Sort/control descriptions: wild-type, EcN harboring wild-type StlA, $n = 12$ biological replicates normalized to the average of the 12 wild-type production values; mock, library sorted based on cell size and doublet exclusion without GFP-based gating, $n = 90$ independent colonies from the sorted pool; top 1%, selection of top 1% brightest cells also gated based on cell size and doublet exclusion, $n = 90$ independent colonies from the sorted pool; mock-mock, mock population retransformed into fresh EcN background with sensor and mock-sorted a second time without GFP selection, $n = 90$ independent colonies from the sorted pool; top 1%-top 1%, top 1% population retransformed into fresh EcN background with sensor and sorted for its top 1% brightest cells, $n = 90$ independent colonies from the sorted pool. *P* values were calculated using Welch's ANOVA test with Dunnett's T3 multiple comparisons test.

is described in more detail below, was achieved using a microfluidic-based workflow (Fig. 1c; mock vs top 1% $p < 0.0001$, mock-mock vs top 1%-top 1% $p < 0.0001$, and additional enrichment from top 1% to top 1%-top 1% $p < 0.0001$, Welch's analysis of variance (ANOVA) test with Dunnett's T3 multiple comparisons test).

**Pop 'n' sort: ultra-high-throughput screening with diffusion mitigation.** Crosstalk is a common problem in the deployment of selection technologies to isolate improved production phenotypes

for metabolites that diffuse or are transported across the cell membrane, including TCA. By diluting cultures and reducing cultivation timescales to minimize accumulation of TCA in the bulk medium, Flachbart et al.[18] were able to isolate enzyme variants with improved PAL activity from a $2.3 \times 10^6$-member error-prone library using a sensor system based on the native TCA-responsive activator HcaR from *E. coli*. Another approach to minimize diffusion of small molecules is to encapsulate single variants microfluidically in fluorinated oil or in gel beads[19,28,29] and to subsequently sort the droplets[30], but this comes at a cost to

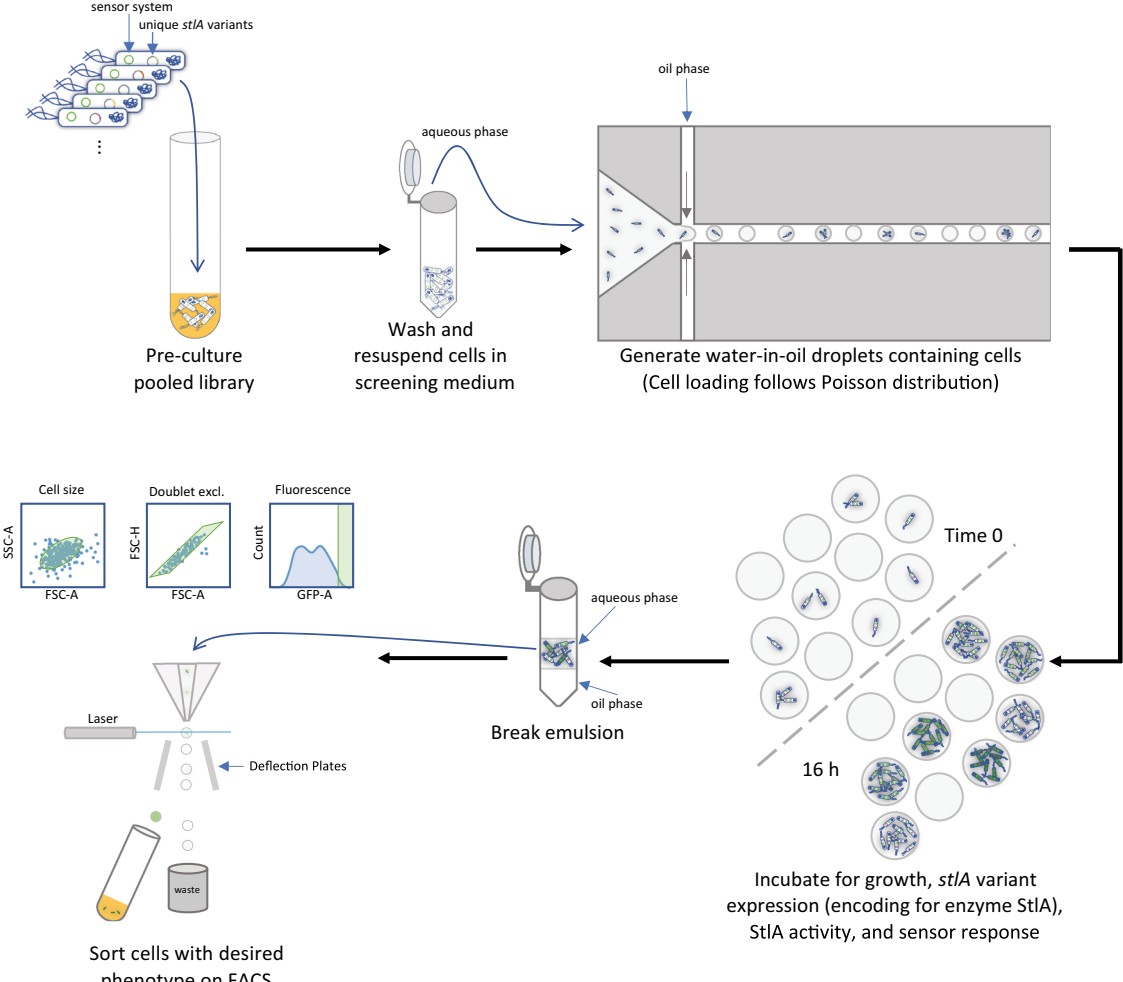

**Fig. 2 Depiction of pop 'n' sort methodology.** EcN cells harboring the high-copy sensor system plasmid (engineered aTF expression, with aTF-repressed *gfp* gene encoding for green fluorescent protein GFP) and a library of low-copy inducible *stlA* variant expression plasmids were grown together within a pool; different cells in this pool contain unique *stlA* sequences. Following a pre-culture step in a rich medium, the cells are washed and resuspended in an adapted M9 minimal glucose medium for sensor screening, which simultaneously induces *stlA* expression and provides the substrate Phe. At this stage, the cells are diluted to an $OD_{600}$ that will result in ~1 cell/droplet loading. These diluted, washed cells in medium serve as the aqueous phase for droplet generation in a fluorinated oil with fluorosurfactant. Because cell loading follows a Poisson distribution, some droplets will be empty and some will contain multiple cells/genotypes at time 0, as shown above. The loaded droplets are incubated until saturation, ~16 h, at which point there are many cells per droplet, and these cells have produced TCA and a per cell GFP signal that is correlated with that TCA production. Since the GFP signal is associated with the cell itself, we can break the emulsions using standard techniques, then sort the cells directly on a FACS for the desired phenotype. See Methods for details on recipes, materials, and protocols.

throughput. While water-in-oil droplet generation can reach frequencies of up to 10,000 droplets per second[30], water-in-oil droplet sorting technologies generally reliably achieve rates of only tens to hundreds of droplets per second[19,30]. This can be increased to approximately a thousand droplets per second by encapsulating the water-in-oil droplets in a second aqueous phase (i.e., water-in-oil-in-water droplets, a.k.a. double emulsions) and sorting on a commercially available FACS[28], but the double emulsion process is technically challenging and prone to instability and settling during sorting.

In the present work, we combine the diffusion mitigation of droplet incubations with the throughput and ease of cell sorting on a FACS in a workflow which we have named pop 'n' sort (Fig. 2)[31], which is enabled by the genotype-phenotype association of our sensor technology. Since the PAL variants were expressed within the same EcN strain harboring the sensor system, the GFP response is linked to the producer cell itself

rather than to the droplet, as is the case in co-cultured systems of producers and dedicated sensor cells[19,29]. Significant enrichments were achieved when using the pop 'n' sort methodology (Supplementary Fig. 2b; mock-mock vs top 1%-top 1% $p = 0.0007$, Welch's ANOVA test with Dunnett's T3 multiple comparisons test) versus when sorting directly from saturated liquid culture (Supplementary Fig. 2a; $p = 0.1408$).

In the pop 'n' sort enrichments described herein, cells were encapsulated microfluidically in water-in-oil droplets at ~1 cell/droplet. The droplets were incubated until the encapsulated cultures became saturated, ~16 h under our media and growth conditions, at which point the emulsions were broken (Methods). The aqueous layer containing the cells was separated from the oil and could be directly sorted on a FACS using standard protocols. The ultra-high-throughput screening approach described above allowed us to screen a more than 1-million member combinatorial library of computationally designed mutations.

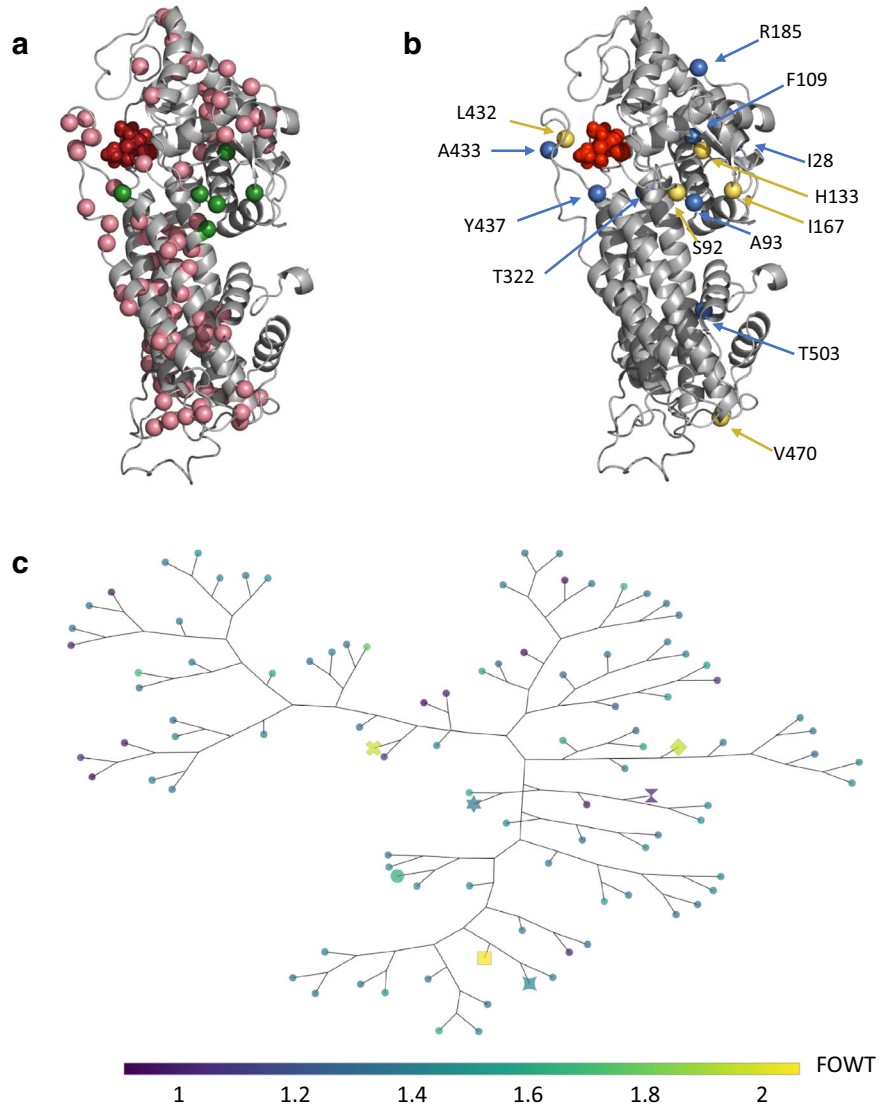

**Fig. 3 Designed and enriched mutation positions.** Positions of interest are highlighted on a homology model generated for StlA using RosettaCM. Active site residues are highlighted in red. **a** Positions targeted for combinatorial mutagenesis during library construction. Any positions mutated in the library templates (see Methods) are highlighted in green, and additional mutations targeted during combinatorial mutagenesis are shown in pink. **b** Positions mutated in top hits. Positions mutated in the final SYNB1934 PAL variant are highlighted in yellow, and those mutated in other top hits (indicated with special markers in **c**, further detail in Supplement) are shown in blue. **c** A phylogenetic tree was generated by computing the distance between all the sequences using the Blosum62 substitution matrix[57] and constructing the tree by neighbor-joining in Biopython[58,59]. Colors indicate activity level normalized to wild-type (FOWT). Markers: square = S92G_H133F_A433S_V470A, diamond = S92G_H133M_I167K_L432I_V470A, x = A93C_H133F_T322W_Y437N, star-square = I28R_S92G_H133F_V470A, large circle = S92G_H133F_R185E, hexagram = S92G_F109A_H133M_T503E, wild-type = hourglass; all other unique variants are shown as a small circle.

**Design and screening of a >1-million member combinatorial PAL library of structure-guided and stabilizing mutations.** A homology model of StlA was constructed using RosettaCM[32] (see Methods) and used to identify active site residues. To improve StlA activity, positions around the active site were targeted for variation. To compensate for instabilities introduced by mutations in and around the active site, phylogenetic (position-specific scoring matrix[33]; PSSM) and co-evolutionary (GREMLIN[34]) analyses were performed to suggest favorable residues (see Methods). As shown in Fig. 3a, the PSSM and GREMLIN evolutionary analyses introduce stabilizing mutations that are widely dispersed across StlA.

Designed mutations were introduced into wild-type StlA and three improved StlA variants (see Supplementary Fig. 3 and surrounding text for origin of templates) using a PCR-based approach as described in the Methods. The constructed plasmid library was transformed into EcN containing our sensor system and prepared for cell sorting using the pop 'n' sort strategy. Following incubation in droplets, cells were sorted by FACS to collect the top 1% brightest cells. The resulting population exhibited an increase in the median TCA production of a 90 clonal variant subsample and was devoid of the majority of low activity variants. The second round of pop 'n' sort isolating the top 1% of the previous top 1% population (top 1%-top 1% from Fig. 1c) further enriched the population to a median of >25% improvement over wild-type with even fewer false positives. The top 1% and top 1%-top 1% pools both showed significant enrichments when compared to their mock-sorted controls ($p < 0.0001$, Welch's ANOVA test with Dunnett's T3 multiple comparisons test).

Following enrichment, ~1500 clones were selected for plate-based assessment of whole-cell activity as described in the Methods, and improved variants were sequenced. Six different PAL variants were of particular interest based on their diverse sequences (Fig. 3c) and activity levels both in initial in vitro assays and in in vitro simulation (IVS) GI model experiments (positions highlighted in Fig. 3b, sequences, and activity in Supplementary Table 1). The phylogenetic tree in Fig. 3c portrays how these variants diverge from wild-type with three to five substitutions. All six variants share a mutation at position 133 from histidine to either phenylalanine or methionine, which increases the packing of the enzyme near the active site and is associated with the increased activity; the histidine of the wild-type enzyme could potentially distort its associated helix by interfering with the carbonyl oxygen from A129. Five of the six top variants also contain an S92G mutation, located at the C-terminal part of the helix, which may serve to increase flexibility by interfering with hydrogen bonding of the carbonyl oxygen. S92G is located near the active site and therefore might influence the binding and release of substrate and/or product. These mutations are combined with diverse additional substitutions with respect to size and charge as indicated by their distances in the tree (Fig. 3c). Further speculations on the potential impacts of the additional mutations can be found in Supplementary Table 2.

**Further characterization of six top PAL variants.** Cells may encounter a variety of pH conditions as they transit the GI tract, and it has been observed that the whole-cell activity of EcN expressing wild-type PAL is strongly dependent on environmental pH (Fig. 4a). We, therefore, sought to determine if cells expressing evolved PAL enzymes displayed a similar activity profile to that of wild-type PAL when subjected to changes in pH. A general trend of reduced overall activity was still observed at lower pH for all variants tested (Supplementary Fig. 4a–d), and activity was improved at more alkaline pH (Supplementary Fig. 4d). Despite an overall reduction in activity for all variants at pH 5, most strains still maintained improvement over wild-type, at least at later time points (Fig. 4b, Supplementary Fig. 4e). Variant A93C_H133F_T322W_Y437N (strain ID EP2495), however, did not perform well under this condition, with an average FOWT < 1 at 4 h (Fig. 4b) and even more severe activity reduction at earlier time points (Supplementary Fig. 4e), so we did not proceed with this variant for further testing.

To assess resumption of strain activity following acid stress, cells were first exposed to medium at pH 5 for 1 h at 37 °C, then assayed for activity at neutral pH (Fig. 4c, Supplementary Fig. 5). In contrast to the very low activity observed when assayed at pH 5 (Supplementary Fig. 4a), cells were able to recover from a 1 h exposure to pH 5 medium and demonstrated similar activity to control cells (Fig. 4c, Supplementary Fig. 5a). All strains retained improvement over the wild-type StlA control after exposure to pH 5 (Fig. 4c, Supplementary Fig. 5b).

Two variants, S92G_H133F_A433S_V470A and S92G_H133M_I167K_L432I_V470A, performed particularly well under both of these pH tests and demonstrated the highest activity at neutral pH. The cell lysate was prepared for EcN harboring each of these two variants and the wild-type PAL control. Following normalization for total protein content, lysates from both variants demonstrated significantly increased $V_{max}$ and $K_M$ values compared to wild-type StlA ($V_{max}$ $p = 0.0058$ and $K_M$ $p = 0.0023$ for EP2315 vs EP2516, $V_{max}$ $p = 0.0135$ and $K_M$ $p = 0.0008$ for EP2315 vs EP2525, based on two-tailed $t$ tests with an unequal variance; Table 1), but there was no significant difference between the two variants with regards to $V_{max}$ or $K_M$.

**Construction and testing of an improved live biotherapeutic strain, SYNB1934.** Though there was no discernable difference between the two lead PAL variants characterized, the S92G_H133M_I167K_L432I_V470A variant, henceforth termed mPAL (for mutant PAL), was selected to construct SYNB1934, an EcN-based live biotherapeutic strain suitable for human dosing[9]. This variant was chosen over S92G_H133F_A433S_V470A because it did not incorporate an additional Phe that could ultimately contribute to dietary Phe load in dosed subjects if cells lysed during transit. In addition to the S92G and H133M mutations discussed above, the variant includes I167K, L432I, and V470A. For reasons discussed in more detail in Supplementary Table 2, we suspect that L432I affects binding/release of the substrate, whereas I167K and V470A likely play a role in stabilizing the enzyme. This strain contains multiple copies of the gene encoding mPAL integrated into the chromosome and, for biocontainment purposes, a mutation of the *dapA* gene to generate auxotrophy for the essential cell wall component diaminopimelate (Fig. 5a). SYNB1934 was also engineered to express the same ancillary Phe-degrading components of SYNB1618, namely the high-affinity Phe transporter, PheP, and the alternative Phe-degrading enzyme LAAD[10]. As is the case for SYNB1618, all antibiotic resistance genes used during the engineering of SYNB1934 were removed upon completion of strain construction.

SYNB1934 was grown and induced for Phe degradation activity in a fully controlled fermenter system to high cell density followed by washing, concentration, reformulation, and lyophilization. This process simulates the production of a drug products intended for human dosing. Resuspended material from both lyophilized SYNB1934 and SYNB1618 demonstrated high viability as measured by fluorescent dye exclusion assay (Supplementary Table 3). In in vitro GI simulations, resuspended SYNB1934 demonstrated a significant increase in the rate of TCA production of approximately two-fold compared with SYNB1618 (Fig. 5b; one-way ANOVA followed by Tukey's multiple comparison test, $p < 0.001$).

To compare in vivo activity of SYNB1934 to SYNB1618, we next moved to NHP studies. We have previously reported that plasma TCA can be used as a unique strain-specific biomarker to track PAL activity of engineered strains; administration of wild-type EcN did not result in the detection of this metabolite[10]. Following an oral bolus of peptides (Peptone) and deuterated Phe (d$_5$-Phe), animals that received SYNB1934 demonstrated a significant increase in plasma exposure to both TCA and d$_5$-TCA compared with animals that received SYNB1618 ($5.4 \pm 1.0$ vs $2.9 \pm 0.9$ mM*h, respectively, for TCA, and $8.7 \pm 2.0$ vs $4.6 \pm 2.0$ mM*h, respectively, for d$_5$-TCA; s.e.; Fig. 5c). In addition, the TCA produced by strains in vivo is quantitatively converted by liver enzymes to hippuric acid (HA) and targeted for urinary excretion[10,35]. Unlike plasma TCA, urinary HA is a common metabolite found in primate urine resulting in significant background levels[10,35,36]. However, the use of oral d$_5$-Phe as a metabolic tracer allowed the calculation of urinary HA excretion specifically attributable to engineered strains. That is because, although there are many aromatic compounds that can be metabolized to HA and targeted for excretion, Phe is not one of them[10], and thus the only metabolic route from oral d$_5$-Phe to urinary d$_5$-HA is through the d$_5$-TCA intermediate produced by PAL. At 6 h post-dosing, urinary d$_5$-HA concentration in animals that received SYNB1934 was more than two-fold higher than that of animals that received SYNB1618 ($534.4 \pm 107.0$ vs $249.1 \pm 25.1$ µg d$_5$-HA/mg creatinine, s.e.; Fig. 5d).

**Discussion**

PKU is typically managed by dietary restriction, which is a significant burden for patients. In many cases, poor dietary

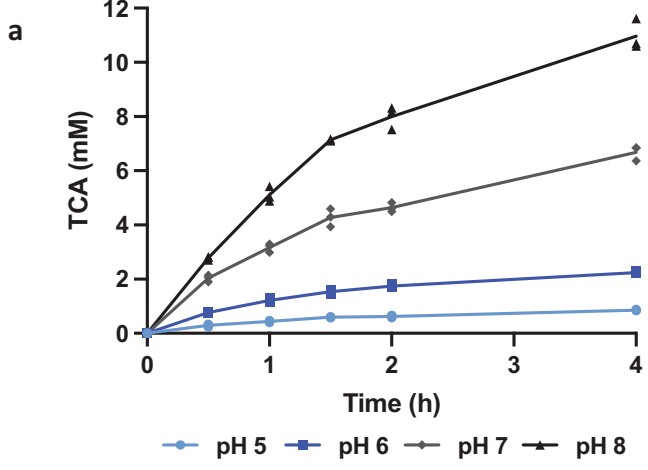

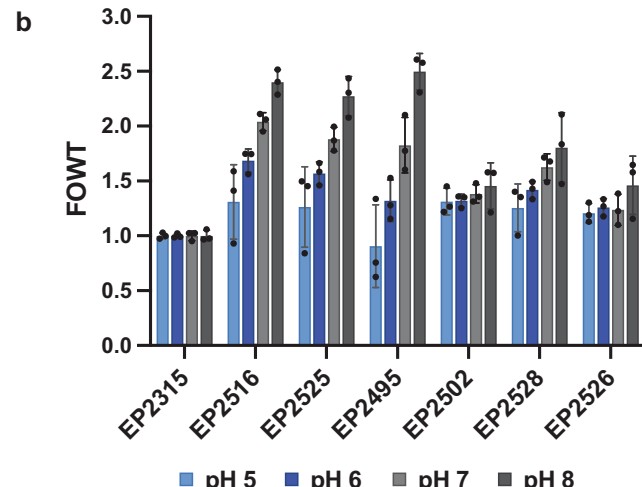

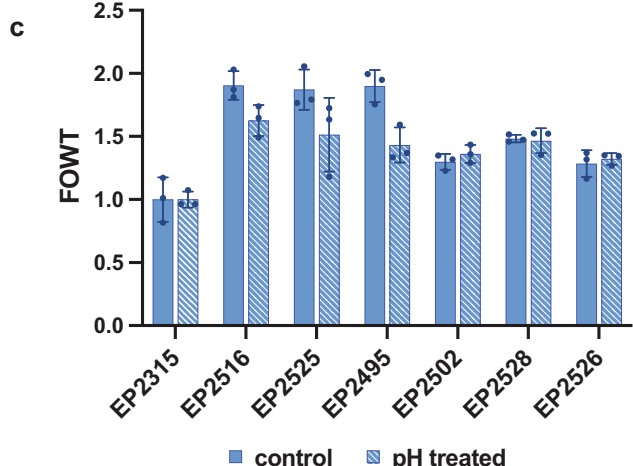

**Fig. 4 Activity at different pH or after exposure to low pH. a** Whole-cell TCA production for EcN with wild-type StlA after 0.5 h, 1 h, 1.5 h, 2 h, or 4 h at pH 5, 6, 7, or 8. $n = 3$ biological replicates with individual data points shown. **b** Whole-cell PAL activity after 4 h at pH 5, 6, 7, or 8, normalized to wild-type StlA activity (fold over wild-type, FOWT). $n = 3$ biological replicates ± s.d. **c** Whole-cell PAL activity after 4 h at pH 7, assayed without pH treatment (control) or after incubation for 1 h at pH 5 (pH treated), normalized to wild-type StlA activity (FOWT). $n = 3$ biological replicates ± s.d. $X$ axis labels refer to strain IDs, EcN with PAL variants. EP2315: wild-type StlA, EP2516: S92G_H133F_A433S_V470A, EP2525: S92G_H133M_I167K_L432I_V470A, EP2495: A93C_H133F_T322W_Y437N, EP2502: I28R_S92G_H133F_V470A, EP2528: S92G_H133F_R185E, EP2526: S92G_F109A_H133M_T503E. For additional time points and non-normalized TCA production data, refer to Supplementary Figure 4 and 5.

**Table 1 Michaelis-Menten parameters from lysate kinetic data.**

| Strain | StlA variant | $V_{max}$ (µmol TCA/min) | $K_M$ (mM Phe) |
|---|---|---|---|
| EP2315 | wild-type | 3.20 ± 0.63 | 1.69 + 0.37 |
| EP2516 | S92G_H133F_A433S_V470A | 11.96 ± 1.64 | 4.06 + 0.44 |
| EP2525 | S92G_H133M_I167K_L432I_V470A | 10.70 ± 1.89 | 4.40 + 0.27 |

$V_{max}$ and $K_M$ average of $n = 3$ biological replicates ±s.d.

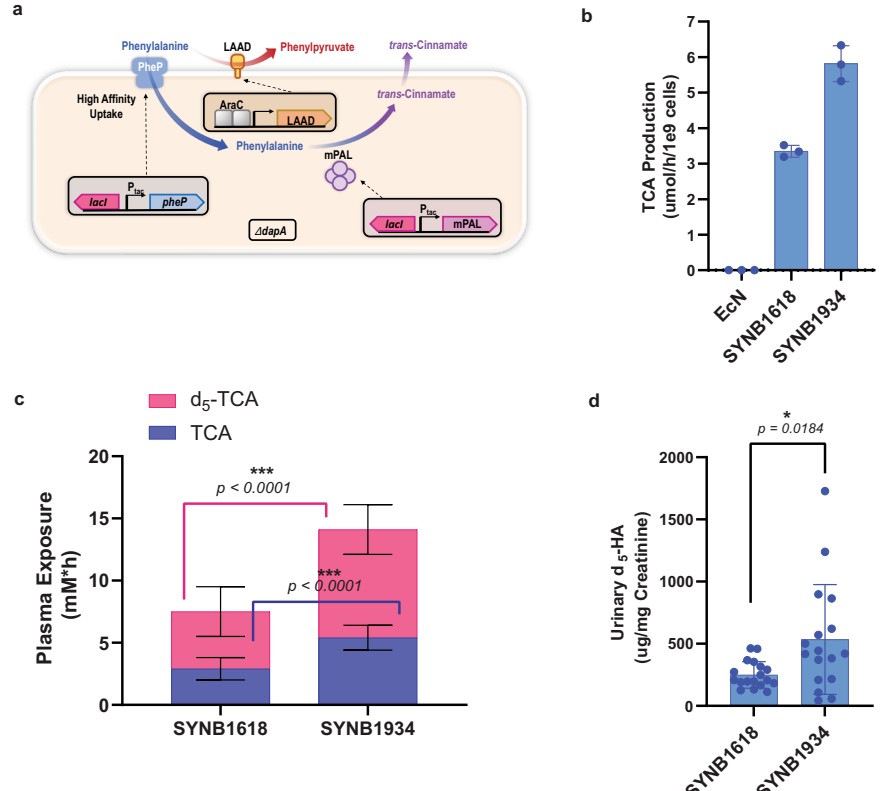

**Fig. 5 In vitro and in vivo activity of SYNB1934.** Optimized PKU strain activity is compared to SYNB1618. **a** Schematic representation of the SYNB1934 bacterium shows key engineered elements, including the genes encoding mPAL, PheP, LAAD, and deletion of the *dapA* gene. **b** TCA production of $2.5 \times 10^9$ resuspended lyophilized SYNB1618 vs SYNB1934 cells was measured in an in vitro simulation of the gut environment. $n = 3$ independent biological triplicates ± s.d. **c** and **d** NHP subjects were dosed orally with a 5 g peptide and 0.25 g $d_5$-Phe bolus followed by dosing with $1 \times 10^{11}$ resuspended lyophilized SYNB1618 or SYNB1934 cells. Plasma areas under the curve (AUCs) for strain-specific biomarkers TCA and $d_5$-TCA are displayed. For **c**, $n = 18$ biologically independent NHP subjects per group ± s.e. For each comparison, data were analyzed using a two-tailed unpaired *t* test ($p < 0.0001$). For urinary $d_5$-HA concentration normalized to creatinine (**d**) $n = 18$ for SYNB1618-treated and 17 for SYNB1934-treated subjects ± s.e (one NHP in the SYNB1934-treated group failed to produce a urine sample over the course of experimentation). Data were analyzed using a two-tailed unpaired *t* test with Welch's correction ($p = 0.0184$).

management can lead to low quality of life. Engineered Synthetic Biotic™ medicines offer the potential for safe, tolerable, reversible, and orally delivered therapies for a variety of metabolic conditions including PKU. EcN was selected as the ideal bacterial chassis for Synthetic Biotic™ strain development. Discovered in 1917, it has been used in human populations for a variety of GI conditions, including inflammatory bowel disease and irritable bowel syndrome[37,38]. EcN has also been reported to interact with the intestinal epithelium to stimulate anti-inflammatory activities and to enhance and maintain barrier function[39]. Importantly, EcN does not exhibit long-term colonization in healthy humans following oral administration[40]. Limitation of bacterial residence time and in vivo replication are attributes that allow predictable and reproducible pharmacologic properties analogous to traditional large and small molecule therapeutics[9]. The inclusion of auxotrophies in engineered strains acts as safeguards to further

ensure predictable outcomes as well as limit growth in the environment following excretion.

This work focused on the application of an engineered aTF biosensor to improve the Phe-degrading enzyme PAL for a clinical PKU treatment strain, followed by the development and preclinical testing of an improved live biotherapeutic, SYNB1934. Biosensor-controlled fluorescence readout allows researchers to phenotype millions of genotypes in a single day[16]. However, crosstalk between high and low producing strains has historically presented difficulties in achieving effective enrichments for improved production[18]. Here, we present a technically simple method, pop 'n' sort, to mitigate diffusion using water-in-oil emulsions for incubation steps, while still sorting individual cells rather than full droplets by FACS. Using the pop 'n' sort methodology and our engineered biosensor, we screened a > 1-million member library for improved whole-cell PAL activity and

identified variants with improved activity. The mPAL that was used in the improved therapeutic strain contained five mutations relative to the wild-type enzyme, three of which are targeted near the active site and we expect to play a role in the binding/release of the substrate and two of which likely have a stabilizing effect.

In in vitro cell lysate experiments, mPAL showed significantly increased $V_{max}$ as well as $K_M$. To mimic expression and activity in the final LBP as closely as possible, protein tags were not included in the construction of enzyme libraries. Although the kinetic analysis on cell lysate does not allow for the decoupling of $k_{cat}$ from enzyme concentration in the reported $V_{max}$ values (see Table 1), normalization to total protein content is more representative of how a head-to-head comparison of therapeutic strains would be performed. Of note, a higher $K_M$ is consistent with a hypothesis throughout the project that a low $K_M$ (higher binding affinity of the substrate to an enzyme) may correlate with more severe inhibition by the product TCA, which is structurally very similar to the substrate Phe. Though the characterization of inhibition in wild-type StlA and mPAL was not undertaken in this work, feedback inhibition by TCA is a common trait in many other kinetically characterized PALs[41–44] and is hypothesized for StlA based on the increased activity observed in cell lysate compared to whole cells (Supplementary Fig. 1).

Exposure to varying pH will undoubtedly be encountered by strains transiting the human GI tract. The effect this may have on strain activity is one parameter that has recently been mechanistically modeled for SYNB1618[45], and future work will aim to perform similar modeling for SYNB1934. All PAL variants identified in this work displayed broadly similar patterns of whole-cell activity in response to changes in environmental pH, although most improved variants still demonstrated increased activity above wild-type PAL under the same conditions. PAL enzymes, in general, are reported to have pH optima in the basic range[43,46–48], but we presumed E. coli would recover and maintain its cytosolic pH near neutral regardless of the environmental pH range tested during these studies[49]. However, increased membrane permeability and the cytosolic accumulation of organic acids such as TCA would be expected as pH decreases, which could result in increased feedback inhibition of PAL. Future work will aim to elucidate the cause of the reduced whole-cell PAL activity observed at low pH, which may warrant the development of a sensor application compatible with ultra-high-throughput screening in acidic conditions. The inability of the screening undertaken in this work, performed at neutral pH, to identify variants with vastly improved pH tolerance is in line with what is frequently referred to as the first law of directed evolution: "you get what you screen for"[50]. Screens can be adapted accordingly to enrich for characteristics of interest, such as activity at different pH, temperature, osmotic strength, etc. That being said, there were some subtle phenotypic differences in the response of lead PAL variant strains to pH changes. Variants were observed that had more drastically reduced activity than the wild-type enzyme at low pH, or that exhibited an increased rate of change in activity over wild-type as pH increased. Other variants displayed the same fold improvement in activity over the wild-type enzyme regardless of environmental pH. These results underscore the diversity contained within the pooled >1-million member mutant library, which, when coupled to the appropriate screen/sensor, can be rapidly redeployed for the selection of alternative desirable enzyme properties.

In a Phase 1/2a clinical trial, SYNB1618 was shown to consume Phe and convert it to the expected non-toxic metabolites in a dose-responsive manner within the human gut of both healthy volunteers and PKU patients[15]. In this study, SYNB1618 was not associated with systemic toxicity, and all subjects cleared the bacteria within days of the last dose. Though not powered to detect a change

in plasma Phe due to the small size of the PKU patient cohort, the results supported the further study of SYNB1618 in Phase 2 clinical trial comprised of a larger cohort of PKU patients (SynPheny-1; NCT04534842). In parallel, SYNB1934 was constructed and tested preclinically, where we have shown that it outperforms SYNB1618 in vitro and in vivo. A Phase 1 study of SYNB1934 was recently initiated (NCT04984525). Relative to SYNB1618, SYNB1934 may offer improved plasma Phe lowering, the potential for lower dosing, or other therapeutic benefits.

## Methods

**Plasmids.** For biosensor screening and PAL variant characterization, the gene encoding StlA from *Photorhabdus luminescens*[11] and StlA variants were expressed from anhydrotetracycline (aTc)-inducible plasmids containing low-copy origins of replication (pSC101). These plasmids encoded an autoregulated TetR and a constitutive spectinomycin resistance cassette; 100 μg/mL spectinomycin was provided in all growth steps for StlA plasmid maintenance. Variant library design and build approaches are described below. The sensor system (engineered TCA biosensor and biosensor-regulated expression of *gfp*) was contained on a single high-copy plasmid with a constitutive ampicillin resistance marker; 100 μg/mL carbenicillin was provided in all growth steps for sensor plasmid maintenance.

**Chromosomally engineered strain construction.** *Escherichia coli* Nissle 1917 (EcN) was purchased from the German Collection of Microorganisms and Cell Cultures (DSMZ Braunschweig, *E. coli* DSM 6601). Phe-degrading clinical candidate strains were constructed through the insertion of genes into the EcN chromosome. SYNB1618 construction has been described previously[10]. The intergenic loci, *malEK, araBC, yicS/nepI, agaI/rsmI, exo/cea,* and *rhtBC* were all identified as suitable insertion sites. These intergenic regions consist of divergent promoters or convergent open reading frames separated by a significant length of DNA such that any inserted sequences would not be expected to lead to polar effects on neighboring genes or promoters. Chromosomal insertions into the EcN genome were performed using the well-characterized lambda Red recombineering approach[51]. For each insertion, (1) pKD3 or pKD4-based plasmid containing 1000 bp of 5′ and 3′ EcN genome homology for recombination was built, followed by (2) insertion of the gene/promoter of interest into the plasmid by the isothermal assembly (HiFI DNA Assembly Master Mix, NEB), (3) amplification of the insertion fragment from the plasmid by PCR (including EcN homology regions and a flippase recognition target (frt) site flanked chloramphenicol or kanamycin resistance cassette for subsequent antibiotic cassette removal, Q5 High-Fidelity Master Mix, NEB), (4) recombineering of the insertion fragment by electroporation via pKD46 and subsequent pKD46 removal, and (5) the removal of antibiotic resistance cassettes via pCP20 and subsequent pCP20 removal. All DNA sequences for genomic insertions used in the construction of SYNB1618 and SYNB1934 are available upon request.

For deletion of the *dapA* gene, two rounds of PCR were performed using nested primers. For the first round of PCR, pKD3 was used as the template DNA. The primers were designed to generate a dsDNA fragment that contained homology adjacent to the *dapA* gene locus in the EcN chromosome and a chloramphenicol resistance gene flanked by frt sites. The primers used in the second round of PCR used the PCR product of the first round as template DNA. EcN containing pKD46 was transformed with the *dapA* knockout fragment by electroporation. Colonies were selected on LB agar containing chloramphenicol (30 μg/ml) and diaminopimelate (Sigma, D1377; 100 μg/ml).

All electroporation was performed in an Eppendorf Eporator (1.8-kV pulse, 1-mm gap length electro-cuvettes). Transformed cells were selected as colonies on LB agar (Sigma, L2897) containing carbenicillin at 100 μg/ml of kanamycin at 50 μg/ml where appropriate.

**Assay media.** Phenylalanine was obtained from VWR (catalog number 97062-556) or Sigma (catalog number P2126), and M9 salts were purchased from Fisher (catalog number DF0485-17). Activated biomass was assayed for whole-cell activity in M9 (6.8 g/L Na$_2$HPO$_4$ + 3 g/L KH$_2$PO$_4$ + 0.5 g/L NaCl + 1 g/L NH$_4$Cl) + 0.5% glucose + 40 mM phenylalanine. An adapted minimal M9 glucose recipe with reduced phenylalanine concentration was used for sensor-based screens: 1% glucose + 13.6 g/L Na$_2$HPO$_4$ + 6 g/L KH$_2$PO$_4$ + 1 g/L NaCl + 2 g/L NH$_4$Cl + 1% LB Lennox + trace metals (0.0002% C$_6$H$_8$FeNO$_7$, 1.8 mg/L ZnSO$_4$·7 H$_2$O, 1.8 mg/L CuSO$_4$·5 H$_2$O, 1.2 mg/L MnSO$_4$·H$_2$O, 1.8 mg/L CoCl$_2$·6 H$_2$O) + antibiotic for plasmid maintenance + 200 ng/mL aTc for *stlA* variant expression + 10 mM L-phenylalanine as substrate.

**Biosensor engineering.** A specific TCA-responsive aTF biosensor was engineered using Zymergen's proprietary sensor development platform. It was enriched from a library of designed sensors by selecting for high per cell GFP in the presence of TCA and low background per cell GFP in the absence of TCA on a FACS. Top biosensor candidates were further characterized for their specificity for TCA over

related ligands in the production host EcN, as well as their correlations to produced TCA using StlA variants of known activity levels (Fig. 1b).

**Homology model generation.** A homology model of StlA was constructed using RosettaCM[32]. Templates were identified by using HHblits and HHsearch[52] searching the Pfam and PDB70 databases. The 10 highest scoring PDB templates were then used by RosettaScripts[53] with the beta energy function[54] to construct an ensemble of homology structures; 200 decoys were generated, and the lowest energy was selected. The code used in this analysis, along with instructions, can be found in the following GitHub repository: https://github.com/Zymergen/AdolfsenIsabella_NatComm.

**Library design.** To design libraries for the target enzymes, we used three approaches: phylogenetic analysis, co-evolutionary analysis, and structural analysis.

To perform the phylogenetic analysis, a PSSM was generated using PSI-BLAST[33] (typically using the default settings of $e$ value = 0.01 and iterations = 3) and used to evaluate single substitutions of the wild-type StlA sequence. The resulting matrix contains a score for every possible amino acid at every position in the protein, corresponding to the amino acid's frequency at that position in the set of homologous sequences returned by PSI-BLAST. These potential substitutions were compared to the score for the wild-type residue at that position and, if significantly higher-scoring substitutions (typical cutoff: $z$ score >2) were available from the PSSM, the amino acid at that given position was selected for experimental verification.

To expand the analysis to include residue couplings, a coevolution analysis was performed for StlA using GREMLIN[34], and all couplings were analyzed and optimized. For positions where the coupling was not optimal according to the analysis, the most optimal amino acid substitution was selected to be tested experimentally.

At last, to target the active site, positions near the active site were identified using the homology model (model generation described in the previous section). If those positions had been observed in the phylogenetic analysis (PSSM score >0), they were included in the library for experimental validation.

**Library build.** The library was built into four templates: wild-type StlA and three variants that had demonstrated improved performance in early engineering efforts (H133M_I167K_V207I, S92G_H133F_Y437N, and A93C_H133M_I167K, all of which had 20–40% improvement in activity over wild-type StlA). The StlA variant templates, each harbored within a low-copy aTc-inducible plasmid backbone, were mixed in an equimolar ratio to provide the plasmid template for an oligo-based library build approach. A separate primer pair (Supplementary Data) was designed to introduce each of the target mutations through a one-piece Golden Gate reaction, using inverse PCR[55] to amplify the entire plasmid at the target mutation site. For the 130 target mutations, 130 separate PCRs (Q5 High-Fidelity Master Mix, NEB) were performed. A subsample of each reaction was run on an agarose gel to allow for quantification of relative band intensity, which was used to normalize and pool together the 130 PCR products. The pooled products were digested by DpnI and gel purified, then circularized using NEB Golden Gate mix (BsaI-v2) at 37 °C for 1 h followed by 60 °C heat inactivation for 5 min. The pooled circularized library of 130 mutations ×4 templates was transformed into NEB10β electrocompetent cells using NEB's protocol. Recovered cultures were transferred into LB with antibiotics for overnight selection of transformants, and a subsample was plated on LB agar plates with antibiotics to confirm adequate library coverage. This process was repeated two more times, generating combinatorial libraries of one and two combinations of designed mutations after a second cycle, and one, two, and three combinations of designed mutations after a third. For the second and third cloning cycles, library plasmid was purified from the overnight selection culture and used as the template in the subsequent round of PCR mutagenesis. Following three cycles of cloning, the low-copy aTc-inducible >1-million member library was transformed into a wild-type *E. coli* Nissle (EcN) host strain containing the sensor system on a high-copy plasmid.

**Droplet-based incubation.** To reduce crosstalk during TCA production and sensor response, cells were encapsulated in microfluidically generated water-in-oil droplets. Libraries were inoculated into 10 mL LB+ antibiotic (refer to "Plasmids" above) in 25 mm test tubes from a sufficient volume of −80 °C glycerol storage stock to maintain library complexity. The cultures were incubated overnight ~16 h to saturation, at which point 1 mL of the culture was centrifuged at 17,800 × *g* for 1 min and the pellet resuspended in an equal volume of filtered sensor response medium (see "Assay media" above). The OD$_{600}$ of the washed cells was measured on a Genesys 20 spectrophotometer with 20× dilution, and the cells were diluted to an OD$_{600}$ of 0.03 in the same medium. An OD$_{600}$ of 0.03 results in about one cell being encapsulated per 40 μm droplet on average at time 0.

Droplets were generated in Sphere Fluidics oil (Pico-Surf™ 1, 2% in Novec™ 7500, Lot # 031117-1 - diluted to 1% surfactant in additional Novec™ 7500) at a rate of ~3.4 kHz in a flow-focusing microfluidic device, fabricated in-house as PDMS chips on glass. Droplets were collected in 1.5 mL Eppendorf tubes and incubated at 33 °C in a standing incubator without tumbling for ~16 h. Following incubation, emulsions were broken by adding a sufficient volume of 1H, 1H, 2H,

2H-Perfluoro-1-octanol (Sigma), vortexing for ~15 s, and pulse spinning ~1 s in a VWR Galaxy minister centrifuge to separate the organic and aqueous layers. The aqueous layer contains the cells which have been released from the droplets. The aqueous phase from the broken emulsions was diluted 100× into filtered phosphate-buffered saline (PBS) prior to cell sorting.

**Fluorescence-activated cell sorting.** Cells were sorted on a Bio-Rad S3E Cell Sorter at a target event rate of 1500 events/s with the sample and collection chambers held at room temperature using ProSort™ Software Version 1.6.0.12. Events were collected in 5 mL polypropylene snap cap tubes (Falcon) containing an initial volume of 0.5 mL LB. Events were gated using three metrics: an elliptical SSC-Area vs FSC-Area gate to isolate EcN cells of a similar size, an FSC-Height vs FSC-Area gate to exclude any doublets, and a GFP-Area (FITC-A) gate to select for cells based on strong sensor response. This gating strategy is exemplified in Supplementary Fig. 6. The sorted volume was diluted into sufficient LB Lennox + antibiotic for overnight recovery at 33 °C 220 rpm. To avoid concerns about genetic drift from multiple outgrowths, the StlA plasmids from recovered populations were purified and transformed into fresh EcN containing the sensor plasmid prior to performing subsequent sorts.

**Plate-based activated biomass assay for TCA production from strains.** 150 μL LB Lennox + antibiotic in standard 96-well plates (VWR catalog number 82050-772) was inoculated from colonies of *E. coli* Nissle containing StlA variant expression plasmids and sensor plasmid. These pre-culture plates were incubated overnight at 33 °C 750 rpm in an Incu-Mixer MP incubator (Benchmark Scientific), covered in VWR porous film (VWR catalog number 60941-086).

The overnight pre-cultures were inoculated 1:100 into 1 mL LB Lennox + antibiotic in deep-well 96-well plates (VWR catalog number 89047-264). The plates were covered in VWR porous film and incubated in an Incu-Mixer at 33 °C 1500 rpm for 2 h into exponential phase. After 2 h, all wells were induced with 200 ng/mL aTc (final concentration, VWR catalog number 200002-828) and placed back in the Incu-Mixer for 4 h at 33 °C 1500 rpm, again covered in VWR porous film. All remaining steps were performed at 4 °C/on ice. After 4 h, the plates were centrifuged at 3214 × *g* for 10 min. The supernatant was decanted, and the pellets were washed in 250 μL cold PBS. The plate was again centrifuged at 3214 × *g* for 10 min and decanted. Pellets were resuspended in 30 μL cold PBS (~40 μL final with cell pellet). To this, 40 μL cold 50% glycerol was added, leading to an OD$_{600}$~25 preparation. The cell preparations were transferred to chilled 96-well PCR plates, covered in foil, and stored at −80 °C until the assay.

For the assay, activated biomass preparation 96-well PCR plate aliquots were thawed at 4 °C, and 4 μL of each OD$_{600}$ = 25 activated biomass was aliquoted across a 96-well PCR plate and stored on ice until assay initiation. To initiate the assay, 96 μL of pre-warmed (37 °C) assay buffer (M9 0.5% glucose 40 mM phenylalanine) was added to the 4 μL activated biomass, mixed, covered in foil, and placed in the 37 °C static incubator. After 4 h, the cells were pelleted by centrifugation at 3214 × *g* 4 °C for 10 min. After pelleting, TCA was quantified from supernatants in a Synergy H1 microplate reader at absorbance 290 nm in UV-Star microplates (Greiner catalog number 655801) following dilution in water to within the linear range of the instrument. The final volume in the plate was 150 μL. A standard curve was used to translate A$_{290}$ measurements to TCA (mM) using trans-cinnamic acid purchased from Sigma (catalog number C80857). TCA concentrations produced from select variants were confirmed by high-performance liquid chromatography. For fold over wild-type (FOWT) normalizations, variant TCA levels were divided by the averaged wild-type TCA values from the same assay.

**Activated biomass assay at various pH or after exposure to low pH.** During further characterization of top PAL variants, activated biomass was more meticulously normalized to OD$_{600}$ = 1 during activity assays. Culture tubes with 10 mL LB Lennox + antibiotic were inoculated 1:100 from 3 mL LB Lennox + antibiotic 33 °C 220 rpm overnight pre-cultures. These were incubated for 2 h at 33 °C 220 rpm, at which point they were induced with 200 ng/mL aTc and placed back in the 33 °C 220 rpm incubator. All remaining preparation steps were done on ice/at 4 °C. After 4 h of induction, the cultures were transferred to 15 mL conical tubes (Falcon) and centrifuged at 3214 × *g* for 10 min, resuspended in 1 mL cold PBS, and transferred to 1.5 mL Eppendorf tubes. They were then washed one more time in 1 mL cold PBS (17,800 × *g* 1 min in a microcentrifuge), and resuspended a final time in 300 μL PBS. OD$_{600}$ measurements were performed in cuvettes, and the cells were normalized to OD$_{600}$ = 50 in 300 μL cold PBS. To the 300 μL OD$_{600}$ = 50 in PBS samples, 300 μL cold 50% glycerol was added for a final OD$_{600}$ = 25. Aliquots were distributed across PCR tubes for each strain and stored at −80 °C until the activated biomass assay could be performed.

For the assays performed at various pH (Fig. 4b), activated biomass was thawed at 4 °C and diluted to OD$_{600}$ = 1 in M9 0.5% glucose 40 mM Phe titrated to pH 5, 6, 7, or 8 in 96-well PCR plates. The plate was covered in foil and incubated at 37 °C for 4 h. The PCR plates were then centrifuged at 3214 × *g* 4 °C for 10 min and the supernatants were quantified by absorbance at 290 nm as described in "Plate-based activated biomass assay for TCA production from strains" above.

For assays performed after recovery from low pH (Fig. 4c), the activated biomass aliquots were thawed and diluted to $OD_{600} = 1$ in M9 0.5% glucose at pH 5 (no Phe) in 96-well PCR plates. The plates were incubated at 37 °C for 1 h without shaking, centrifuged at 3214 × g 4 °C for 10 min and washed in PBS, and then were assayed for activated biomass activity at neutral pH as described in "Plate-based activated biomass assay for TCA production from strains" above. To control for cell loss during wash steps to remove pH 5 medium, fresh activated biomass was added to 96-well PCR plates, washed alongside the samples incubated at pH 5, and then assayed for activated biomass activity; these samples are labeled "control" in Fig. 4c. For FOWT normalizations, variant TCA levels were divided by the averaged wild-type TCA values from the same assay.

**Comparison of whole cells vs lysates for examination of PAL expression, abundance, and activity.** A series of PAL-expressing strains were constructed in EcN, which included strains (2) with a single chromosomal insertion of stlA at separate locations, a strain containing both chromosomal stlA gene copies in the same background, a strain with a low-copy stlA-expressing plasmid (pSC101 origin), and a strain with a high-copy stlA-expressing plasmid (pUC origin). In each of these strains, the same stlA coding sequence, ribosome binding site, and aTc-inducible promoter were used so that the only difference between strains was the locations of chromosomal insertion and/or stlA gene copy number. For strain growth and PAL induction, 50 mL baffled flasks with 10 mL of LB Lennox broth containing appropriate antibiotics were inoculated 1:100 from overnight cultures. Flasks were grown shaking at 37 °C 250 rpm in an Eppendorf orbital shaker for 2 h to bring cultures into exponential phase, at which point aTc was added at 200 ng/mL aTc (final concentration, VWR catalog number 200002-828). Cells were allowed to grow induced for an additional 4 h before centrifugation for 10 min at 5000 × g and resuspension of pellets in an equal volume of ice-cold PBS.

To measure PAL activity of whole cells, strains were resuspended to an $OD_{600}$ of 0.1 in 1 mL Phe assay buffer (M9 0.5% glucose-containing 40 mM Phe) in microfuge tubes and placed in a 37 °C heat block. Samples were removed every 30 min for 2 h and TCA concentration was determined by $OD_{290}$ measurement as described above. Rates of TCA production were extrapolated from the concentration of TCA measured in the supernatant over time.

To measure PAL activity from lysates, a 6 mL volume of resuspended cells were passed through a Microfluidics™ LV1 Low volume Microfludizer® homogenizer at a pressure of 18,000 psi. Each strain was passed through the homogenizer three times to ensure complete lysis. The resulting lysates were spun down at 12,000 × g for 20 min to pellet insoluble material. Total soluble protein concentration for each cleared lysate was determined via BCA assay (Pierce, catalog number 23227). Rates of TCA production were determined similar to the method used for whole cells described above, with the exception that 30 mg of soluble protein was used to provide PAL for the assay rather than resuspension of cells.

Sodium dodecyl sulfate–polyacrylamide gel electrophoresis (SDS–PAGE) gels were run using whole cells of WT EcN and the engineered strain series described above. Cells were grown and induced as described. Whole cells of each induced strain were resuspended to an $OD_{600}$ of 0.5 and boiled in 1× lithium dodecyl sulfate sample loading buffer for 15 min (Invitrogen, catalog number NP0007). A 20 μL volume of each sample was loaded onto 4–12% bis-tris stacking SDS–PAGE gel (Invitrogen, catalog number NP0322). For standards, iBright Prestained Protein Ladder (Invitrogen, catalog number LC5615) was included to estimate protein size. Gels were run in MES SDS Buffer (Invitrogen, catalog number NP0002) at 200 V for 30 min and stained with SimplyBlue SafeStain (Invitrogen, catalog number LC6065) according to the manufacturer's instructions.

**Michaelis–Menten kinetic parameters determined from cell lysate.** Activated biomass was prepared from 10 mL culture tubes as described in the section above, and then lysed for kinetic parameter determination. To prepare lysate, thawed biomass samples were diluted and sonicated using a Branson Digital Sonifier with microtip, then the soluble fraction of the lysate samples were used for the kinetic assay. Total protein in the lysate samples was measured via Bradford Assay, and all samples were normalized to 10 μg total protein loading per well for the kinetic assay. The lysate samples were incubated in M9 0.5% glucose with Phe concentrations ranging from 40 mM Phe down to 39 μM with twofold dilutions (assay buffer without Phe was also included as a control). The kinetic assay was performed in UV-star 96-well microplates (Greiner) with TCA quantified by $A_{290}$ measurements every minute using a BioTek Synergy H1 microplate reader set to 37 °C static incubation. Michaelis–Menten model fitting was performed using a nonlinear regression (nls function in R with formula $V = (V_{max} * [S])/(K_M + [S])$) for the rate data from three batch replicates. Example model fits can be found in Supplementary Fig. 7. The rate $V$ used in the nonlinear regression was calculated from the first hour of activity for each Phe concentration tested, where activity remained linear.

**Bioreactor growth of integrated strains, lyophilization, and determination of viability.** All strains were grown in fermentation media, which was prepared as followed: yeast extract (40 g/L), $K_2HPO4$ (5 g/L), $KH_2PO_4$ (3.5 g/L), $(NH_4)_2HPO_4$ (3.5 g/L), $MgSO_4*7H_2O$ (0.5 g/L), $FeCl_3$ (1.6 mg/L), $CoCl_2*6H_2O$ (0.2 mg/mL), $CuCl_2$ (0.1 mg/L), $ZnCl_2$ (0.2 mg/L), $NaMoO_4$ (0.2 mg/L), $H_3BO_3$ (0.05 mg/L),

Glycerol (25 g/L), Antifoam 204 (125 μL/L). Production of strains began by thawing a vial from a cell bank and culturing 1 mL of the thawed vial in 50 mL fermentation media supplemented with diaminopimelate (300 μg/mL) in a 500 mL Ultra-Yield™ flask (Thomson Instrument Company). Cells were grown with shaking at 375 rpm until an $OD_{600}$ of 10–15 was reached, at which point the cultures were used to inoculate 24.5 L of media in a HyPerforma™ Thermo Scientific 30 L Single-Use Fermenter at a starting $OD_{600}$ of 0.000016. The fermenter was controlled at 37 °C, 60% dissolved oxygen (DO) concentration, and pH 7 using ammonium hydroxide. For SYNB1618, at an $OD_{600}$ of 1.5, cells were activated by the creation of a low oxygen environment (10% DO), and the addition of Isopropyl β-d-1-thiogalactopyranoside (IPTG, 1 mM). A nutrient feed was also started at this time (15 mL/L-h of 271.75 g/L yeast extract, 86.27 g/L glycerol) and continued until the end of fermentation. After 3.5 h from the addition of IPTG, the nutrient feed rate was doubled to 30 mL/L-h. For SYNB1934, at an $OD_{600}$ of ~1.5, cells were activated by the addition of IPTG (1 mM), and DO was maintained at 30%. A nutrient feed was also started at this time (11.25 mL/L-h of 271.75 g/L yeast extract, 86.27 g/L glucose) and continued for 3.5 h. After 3.5 h, the nutrient feed rate was doubled to 22.5 mL/L-h for a period of 2.75 h. For both strains, L-arabinose was added to the fermentation (10 mM final concentration) for the final hour of fermentation. Control strain SYN094, which contains no Phe degradation components, was grown in a 5-L bioreactor in fermentation media supplying a steady DO content of 30% until the stationary phase was reached. Strains were harvested by tangential flow filtration (TFF).

At the end of TFF, the supernatant was discarded and cells were resuspended to a final concentration of 150 $OD_{600}$ in lyoprotectant buffer (10% wt/vol Trehalose, 50 mM Tris, pH 7.5). The formulated cell suspension was used to fill 2 mL glass amber vials and lyophilized to a final water content of <5%. Lyophilized material was stored at 4 °C. Dried lyophilized powder was reconstituted with PBS (Quality Biological, 114-056-101) to match the pre-lyophilization volume. This reconstituted cell suspension was used for measuring activity, viability.

To determine bacterial cell viability, resuspended cells were diluted and stained with SYTOX Green nucleic acid stain (Life Technologies). Live and dead stained cells were counted directly on a Nexcelom Bioscience Cellometer X2 image cytometer per manufacturer's protocol.

**IVS of the gut environment assays.** The in vitro gastric simulation model was designed to simulate key aspects of oral administration in humans, including gastric oxygen concentration, pepsin secretion, and gastric pH. The IVS assay is comprised of incubations in 96-well microtiter plate format designed to simulate human stomach conditions[56]. In brief, lyophilized cells were resuspended in PBS at room temperature. Bacterial cell concentrations were determined by counts of viable and/or total cells. Aliquots of cells were resuspended in 0.077 M sodium bicarbonate buffer at $5.0 \times 10^9$ cells per mL. This solution was then mixed with equal parts of simulated gastric fluid[56] containing 20 mM Phe, and incubated for 2 h at 37 °C with shaking in a polycarbonate in vitro hypoxic chamber (Coy Lab Products) calibrated to 2% oxygen. The resulting SYNB1618 cell density in SGF was $2.5 \times 10^9$ cells/mL. To determine PAL activity, SGF aliquots were collected periodically and centrifuged at 5000 × g for 5 mins using a tabletop centrifuge, followed by liquid chromatography-tandem mass spectrometry (LC-MS/MS) quantification of metabolites, including Phe and trans-cinnamate (PBS).

**Studies of strain activity in NHPs.** NHP studies were performed at Charles River Labs (Shrewsbury, MA) in compliance with all applicable sections of the Final Rules of the Animal Welfare Act regulations (Code of Federal Regulations, Title 9), the *Public Health Service Policy on Humane Care and Use of Laboratory Animals* from the Office of Laboratory Animal Welfare, and the *Guide for the Care and Use of Laboratory Animals* from the National Research Council. Twelve male cynomolgus monkeys aged 2–5 years were used (2.5–4 kg), and were maintained on International Certified Primate Chow (PMI nutrition, 5048). Standard operating procedures related to NHP studies have been reviewed and approved by Charles River Laboratories' Institutional Animal Care and Use Committee. All animals in the cohort were in a good health at the beginning of the study and washed out for at least 7 days between studies. Three single-dose studies were performed to compare SYNB1934 with SYN1618. On each of the experimental days, six NHP subjects were dosed with SYNB1618 or SYNB1934 and the data presented above are the combined results of three experiments.

The animals were fasted overnight the day before dosing (Day 1) and throughout the procedures on Day 1 without exceeding a maximum of 24 h. On the morning of Day 1, a baseline blood sample was drawn from each monkey by venipuncture. The animals were temporarily restrained for dose administration, but not sedated. Lyophilized bacteria were resuspended and administered orally at $10^{11}$ Live Cell doses to each animal, together with 5 mL of 0.36 M sodium bicarbonate, 7.7 mL of 20 mg/mL L-phenyl-D5-alanine (C/D/N Isotopes Inc.), and 6.1 mL of 500 g/L Peptone peptic digest (Sigma). Plasma and cumulative urine were collected for further analysis. Following dosing, animals were then returned to their cages, and a clean urine collection pan was placed at the bottom of each cage. In all, 6 h after initial dosing, the cumulative volume of urine was measured and recorded, and urine samples were stored at −80 °C. Blood samples were collected at 0.5, 1, 2, 4, and 6 h post dose, and plasma was prepared and frozen at −80 °C.

Concentrations of metabolites were quantified using a derivatization assay with LC-MS/MS detection.

**LC-MS/MS performance**. Quantification of TCA, d5-TCA, HA, and d5-HA were performed using targeted multiple reaction monitoring (MRM) mode in a Thermo TSQ Quantum Max triple quadrupole Liquid Chromatography-tandem Mass Spectrometry (LC-MS/MS) system. The standards used were *trans*-Cinnamic acid (Acros, 158570050), *trans*-Cinnamic acid-d5 (CDN Isotopes, D-5284), Hippuric acid (Sigma, 112003), and Hippuric acid-d5 (CDN Isotopes, D-5588).

Standards were prepared in water with the following concentrations: 0.032, 0.16, 0.8, 4, 20, 100, and 250 µg/mL. Samples were stored at −80 °C prior to analysis. Urine samples were diluted 40-fold in water prior to sample processing. Creatinine (Sigma, 60275) was added to the standard mixture when analyzing urinary HA and d5-HA (0.32, 1.6, 8, 40, 200, 1000, and 2500 µg/mL). In a 96-well plate, 10 µL of the standards and samples were transferred, followed by the addition of 90 µL derivatization solution (50 mM of 2-hydrazinoquinoline, dipyridyl disulfide, and triphenylphosphine in acetonitrile with 1 µg/mL of isotopically labeled internal standard 13C9-15N-Phe (Cambridge Isotopes, CNLM-575-H-PK) and d5-creatinine (CDN Isotopes, D-7707). The plate was heat-sealed with a ThermASeal foil, mixed, and incubated at 60 °C for 1 h to derivatize the samples. The derivatized samples were then centrifuged at $3200 \times g$ for 5 m. To another plate, 20 µL of the derivatized samples was transferred and further diluted with 180 µL of 0.1% formic acid in water/acetonitrile (140:40). The injection volume used was 10 µL, and the run time was 4.25 m at a flow rate of 0.5 mL/minute. Mobile phase A was 0.1% formic acid in the water, and mobile phase B was 0.1% formic acid in acetonitrile/isopropanol (90:10). Chromatographic separation was carried out using a Phenomenex C18 column (3 µm, 100 × 2 mm) with the following gradient: 10% B from 0 to 0.5 m, 10 to 97% B from 0.5 to 2 m, 97% B from 2 to 4 m, and 10% B from 4 to 4.25 m. Multiple reaction monitoring in positive mode was used for tandem mass spectrometry analysis. The following mass transitions were monitored for quantitation: TCA (290/131), d5-TCA (295/136), HA (321/160), d5-HA (326/160), and creatinine (114/44).

**Statistical analysis**. Group means, standard errors/deviations, and linear regressions were calculated in Microsoft Excel. To calculate *p* values, unpaired student *t* tests, one-way ANOVA followed by Tukey's multiple comparison tests, or Welch's ANOVA with Dunnett's T3 multiple comparison tests were performed in Graphpad Prism or in Excel. Areas under the curve were calculated with the linear-trapezoidal method using Graphpad Prism and baselines were set by the average values at time 0 for each applicable experiment.

**Reporting summary**. Further information on research design is available in the Nature Research Reporting Summary linked to this article.

## Data availability

All data generated or analyzed during this study are included in this published article (and the accompanying supplementary information files). Flow cytometry data (Fig. 1b) are provided as .fcs files in the compressed folder "Fig1b_fcs_files" in the Source data. All other data are provided in the file "Source data.xlsx." *E. coli* Nissle strain 1917 was obtained from DSMZ and used for the construction of engineered strains leading to the generation of SYNB1618 and SYNB1934. All strains described in this manuscript were derived from the same parental background. Engineered strains described in this manuscript can be made available subject to an MTA, which can be requested by contacting the corresponding authors. The complete genome sequence of SYNB1618 is available under BioProject ID: PRJNA482064 (https://www.ncbi.nlm.nih.gov/bioproject/?term=PRJNA482064). The complete genome sequence of SYNB1934 is available under BioProject ID: PRJNA749270 (https://www.ncbi.nlm.nih.gov/bioproject/?term=PRJNA749270). Source data are provided with this paper.

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

## Acknowledgements

The authors thank Caitlin Allen, Mark Charbonneau, Caroline Kurtz, and Dave Hava for feedback throughout this project and during the preparation of this manuscript.

## Author contributions

K.J.A., A.G.L., V.M.I. and A.A. supervised the project. K.J.A., I.C., V.M.I. and A.A. analyzed the data. J.S. conducted sensor discovery. P.G. oversaw enzyme engineering and provided all PAL design recommendations. L.E.F. computationally designed the necessary primers to introduce the mutations, and the combinatorial PAL library was constructed by K.J.A. Sensor-based screening was performed by K.J.A., L.W., and I.C. Plate-based activated biomass assays and variant sequencing was performed by K.J.A., I.C. and C.W. L.E.F. performed PyMOL figure generation. A.G.L., P.G., J.S., and J.H.K. provided oversight. C.M was responsible for candidate strain construction and performance/analysis of in vitro experiments. A.A. and M.M. were responsible for fermentation and formulation of strains. M.J.C. performed and analyzed mass spectroscopy analysis. T.M and L.R. oversaw the performance of NHP studies.

## Competing interests

The authors declare the following competing interests: all authors hold stock in Synlogic, Inc. or Zymergen, Inc. and may gain or lose financially through publication.
