## [Peer Review File · Nature Communications]

Reviewers' Comments:

Reviewer #1:

Remarks to the Author:

The manuscript by Adolfsen, Callihan et al. describes the engineering of an improved bacterial therapeutic for phenylketonuria (PKU). The new bacterial therapeutic has about 2 fold more PAL activity than the previously designed therapeutic. PAL is an enzyme that metabolizes phenylalanine and bacterial therapeutics are being developed to lower phenylalanine levels in the GI-tract. The authors statement that 'there remains a need for safe orally delivered therapeutic treatment options for PKU patients that will address the needs of the entire patient population regardless of age or genetic background' is the correct assessment of the needs of PKU patients. This is a very well written manuscript that overall describes a large number of complex experiments conducted clearly and concisely.

The following recommendations are made to improve the manuscript:

p. 20 line 370

'This work focused on the application of an engineered aTF biosensor to improve the Phe-degrading enzyme PAL for a clinical PKU treatment strain.'

This sentence should be revised, the application of an engineered aTF biosensor is only one aspect of the large amount of work described.

p.2 line 35 'A lead enzyme candidate from this screen was used in the construction of SYN1934, a chromosomally integrated strain containing the additional Phe-metabolizing and biosafety features found in SYN1618.'

AND p.17 line 306

'SYN1934 was also engineered to express the same ancillary Phe degrading components of SYN1618, namely the high affinity Phe transporter, PheP, and the alternative Phe-degrading enzyme L-amino acid deaminase (L-AAD)10.'

This means that SYN1934 produces L-amino acid deaminase that catalyzes the conversion of Phe to phenylpyruvic acid and ammonia and phenylalanine ammonia lyase (PAL) which converts Phe to trans-cinnamic acid and ammonia.

The authors should discuss any potential toxicity from phenylpyruvic acid and ammonia and how they monitored for the potential toxicity and should mention what is known about trans-cinnamic acid toxicity.

p.10 line 203 'To compensate for instabilities introduced by mutations in and around the active site, evolutionary (position specific scoring matrix29) and co-evolutionary (GREMLIN30) analyses were performed to suggest favorable residues. As shown in Fig. 3a, the incorporation of phylogenetic analyses to introduce stabilizing mutations leads to designed mutations that are widely dispersed across StIA.'

And corresponding method section on p.26 line 490 – 502

The phylogenetic analysis and the position specific scoring matrix (PSSM) are not sufficiently explained.

p. 14 line 269 'In contrast to the very low activity observed when assayed at pH 5 ..., cells were able to recover from a 1 h exposure to pH 5 medium ... and demonstrated similar activity to control cells.'

The authors should describe how long the enzyme might be exposed to what pH while passing through the human GI system after oral administration.

p.16 line 291

'Of note, a higher KM is consistent with a hypothesis throughout the project that a low KM may correlate with more severe inhibition by the product TCA.'

This statement does not seem to be consistent with what a low KM value means in enzymology.

p.19 'Fig. 5 In vitro and in vivo activity of SYN1934. Optimized PKU strain activity is compared to SYN1618. a, Schematic representation of SYN1934 chromosome shows key engineered genetic elements, including approximate location of genes encoding mPAL, PheP, LAAD, and deletion of dapA gene.'

Fig 5A rather looks like schematic representation of a bacterium than a bacterial chromosome

Is the schematic representation misleading or the figure legend incorrect?

Reviewer #2:

Remarks to the Author:

Title: Improvement of a synthetic live bacterial therapeutic for phenylketonuria with biosensor-enabled enzyme engineering

Authors: Kristin J. Adolfsen, Isolde Callihan, Catherine E. Monahan, Per Jr. Greisen, James Spoonamore, Munira Momin, Lauren E. Fitch, Mary Joan Castillo, Lindong Weng, Lauren Renaud, Carl Weile, Jay H. Konieczka, Teodelinda Mirabella, Andres Abin-Fuentes, Adam G. Lawrence, Vincent M. Isabella

Recommendation:

Accept with minor revision

Overview and general recommendation:

Phenylketonuria (PKU) is a metabolic disease where patients have genetic defects in resulting in deficient phenylalanine (Phe) metabolism, which can result in neurological impairments if left untreated. The authors have previously developed a novel treatment for PKU in a form of engineered *E. coli* Nissle 1917 (EcN) strain (SYNB1618) expressing Phe metabolizing enzyme (phenylalanine ammonia lyase (PAL)), capable of turning Phe into a non-toxic product trans-cinnamate (TCA). In this work, the author has demonstrated a biosensor-based high-throughput screening method to screen and enrich a pool of engineered EcN with rationally engineered PAL enzymes. The plasmid-based biosensor was designed to detect TCA production from the EcN variants. In the absence of TCA, a sensor molecule bound the DNA operator, preventing the reporter *gfp* to be expressed. EcN variants were captured in microfluidic droplets allowing the biosensor system to detect whole cell PAL activity, followed by the use of fluorescence-activated cell sorter (FACS) to select cells with high GFP expression. After a few rounds of screens and enrichments, top hits were then characterized using two sets of experiments: 1) observing whole cell TCA production at various pH background to select variants that remained active with changing pH, and 2) exposing the cells to low pH and later observed the TCA production to select variants that were able to recover from a low pH environment. The top two hits were then characterized for their Michaelis-Menten parameters via lysate kinetic experiment, and one final PAL hit was selected to generate EcN-derived SYNB1934 strain. The strain was then compared with SYNB1618 in simulated gastric model and in non-human primate studies. The authors found SYNB1934's PAL activity to be about twice as effective as that of SYNB1618 in these models.

Overall, this manuscript is well written, and the data provided are sufficient to support the flow of the story and the conclusions the authors made. The novel methods and findings would provide valuable insights into the future development of biosensor-based high-throughput screening platform of similar applications as well as the development of therapeutics on PKU using engineered EcN strains. The methodology presented in this work meets the standard practice in the field and is provided with enough details to be reproduced by others. Nevertheless, the following comments should be addressed during revision.

Major comments:

1. The authors reported most of the data in this paper as FOWT "fold over wild-type" which is great for showing method and strain improvements. However, the authors did not show the wild type data themselves along with their variances in most of these plots (exceptions will be discussed in major comment #2). This reduces the transparency in the data because we do not know how consistent the wild-type samples perform in these assays before they are taken the averages and used as benchmarks. I would recommend the authors to include the wild-type data points with correct error bars to all of the FOWT data plots (for the reason that I emphasized the word "correct" error bars, please see major comment #2 as well). I understand that the average

values of the wild-type in these plots would be one because they are used as benchmarks for normalization. However, the error bars should still be reflective of the variance in the wild-type samples in each experiment.

2. The authors did include wild-type samples in certain FOWT plot, namely supplementary figure 4e, f, g, h, and supplementary figure 5c. However, I noticed that all the error bars of the wild-type strain (EP2315) in these figures appear to be completely flat (or zero SD) which is not possible because I observe variances in the raw trace data in supplementary figure 4a, b, c, d and supplementary figure 5a, b. This could be an error during normalization process which left the wild-type data in the FOWT plots with just the average value with no variation between samples. Please restore the correct error bars to the wild type data in these figures to reflect the true variances of the wild-type strain.

3. In supplementary figure 1, the authors showed whole cell and lysate TCA production data with increasing number of PAL copies within the cells, inferring the increasing amount of PAL proteins within the cells or lysate. However, the authors offered no evidence showing that increasing the copy number of PAL increased the amount of PAL protein production. Please include the data showing the increase in PAL production as a function of copy number in a form of SDS-PAGE along with western blot or ELISA if possible.

4. In line 291, the author wrote "Of note, a higher KM is consistent with a hypothesis throughout the project that a low KM may correlate with more severe inhibition by the product TCA." This conveys that the authors theorized product inhibition in these enzymes but did not demonstrate the product inhibition behavior in the Michaelis-Menten plots. In figure 6, we do not observe the curve(s) turning downward due to product inhibition. I would suggest the authors re-run the Michaelis-Menten experiment with higher concentrations of Phe to induce product inhibition and reduction in the rates of reactions of these enzymes.

5. Kinetic parameters determined from cell lysates will not be very accurate due to other protein contents in the cells. Besides potential inhibitors in the cell lysates, the authors cannot determine k_{cat} as they do not know the exact concentrations of the enzymes in these reactions. The V_{max} values are still dependent of the actual enzyme concentrations. I would recommend the authors to purify the proteins prior to the kinetic characterization. However, if these PAL proteins contain no tag for the purpose of being delivered as therapeutics, the kinetic parameters from the tagged versions of these enzymes (for the purpose of purification) would be acceptable and better than the current kinetic parameters from cell lysates.

6. NHP study is missing a negative control where wild-type EcN is being fed to compare with SYN1618 and SYN1934. While missing this control does reduce the robustness of the data in figure 5c and 5d, it is understandable as it is difficult to have a large number of monkeys in a single experiment to explore all of the conditions. Moreover, the authors already ran this EcN control condition comparing to SYN1618 in Isabella V., et al (2018). I would suggest the author mention the negative control and how it underperformed SYN1618 and reference Isabella V., et al (2018) paper in the text.

Minor comments:

1. In supplementary figure 1, please perform one-way ANOVA with multiple comparison (Ordinary ANOVA with Tukey test, assuming equal populations and variances) to convince the readers about the plateau of whole cell data and the increase of TCA production in lysate data with the increase of protein production.

2. The authors did not describe the details of experiments in supplementary figure 1 in the method. Detailed procedure of this specific whole cell and lysate assays should be described for the reader to comprehend the comparison between the two experiments. While the whole cell experiment was described in the method and the readers can assume a similar protocol was ran. There was no specific indication about the timepoint that this data set was taken. Moreover, there was no detailed explanation for this lysate assay in the method session. A similar lysate assay protocol was later described but that was for Michaelis-Menten characterization which included the protein normalization step, which I assume would be different in this experiment. Please describe these procedures in appropriate sub-sections in the method section.

3. Figure 1c, supplementary figure 2 and 3 lack the statistics that conveys the differences among the sorted groups. Please perform Welch ANOVA test with Dunnett T3 multiple comparison test, assuming different N in each group and wide range of variances, to determine significant differences among sorted populations and engineering rounds.

4. Line 48 of supplementary document mentioned "4 wild-type PAL colonies (green), and 4 H133M I167K colonies (red, an early hit with improved activity)". However, I'm only seeing the 4 H133M I167K colonies in green (not in red) on supplementary figure 2 and the wild-type PAL data is missing. I assume the authors reported FOWT and remove the wild-type data after restructuring the data. However, I would still recommend adding the wild-type data back as mentioned in major comment#1 and make sure the color in the caption matches with the actual color in the plot.
5. Line 193 – encapsulated cultures "became" saturated (missing a verb)
6. Please consider rewrite this sentence in line 291 "Of note, a higher KM is consistent with a hypothesis throughout the project that a low KM may correlate with more severe inhibition by the product TCA." It's rather difficult to understand what the authors try to convey.
7. It wasn't clear in the text why one out of the two final hit constructs was chosen over the other to be incorporated into SYNB1934. Please describe the thought process along with relevant supporting information in the beginning of "Construction and testing of an improved live biotherapeutic strain, SYNB1934" section.
8. More details should be provided about the in vitro GI simulation in the text part (Line 315-317). Upon reading the method, it seems this model mimics the gastric. Is it possible to simulate other parts of the gut besides the stomach? I assume the pH and other environment factors would be drastically different from stomach to duodenum, jejunum, and ileum. It would be great to see the activities of the three strains at different parts of simulated gut environments to see the location where the cells would be most active.
9. In figure 5b, please perform one-way ANOVA with multiple comparison (Tukey method) to demonstrate the differences between the three strains in simulated GI model
10. Line 653: the method did not mention the addition of arabinose for induction of LAAD in SYNB1934 strain
11. Line 713: missing a negative sign at "-80"
12. Line 764: "The complete genome sequence of SYNB1934 is available under BioProject ID: XXXXXXX" Please fill out the correct ID.
13. Line 707: the text reads "resuspended and administered PO 1011 Live Cell doses to each animal". Please describe PO abbreviation or change the word to "orally"

All changes to the Main Text and Supplemental Information are tracked throughout. Changes made to manuscript other than Reviewer comments:

- Since first submission, a clinical trial using SYN1934 has commenced. Added second to last sentence, “A Phase 1 study of SYN1934 was recently initiated (NCT04984525).”
- Added “Data Availability” section after Methods stating, “Source data are provided with this paper.” – All source data is included with the resubmission
- Where appropriate, bar graphs with plots that feature information about the distribution of the underlying data were updated. All data points are now shown for plots with a sample size less than 10.
- NHP protocol in the Methods section was updated, it erroneously stated a sample size of 10 NHP subjects per experiment rather than the correct value of 12 (note that the correct values and protocols were included in the Reporting Summary).
- References updated where appropriate
- Spelling/Grammatical errors were corrected throughout

REVIEWER COMMENTS

Reviewer #1 (Remarks to the Author):

The manuscript by Adolfsen, Callihan et al. describes the engineering of an improved bacterial therapeutic for phenylketonuria (PKU). The new bacterial therapeutic has about 2 fold more PAL activity than the previously designed therapeutic. PAL is an enzyme that metabolizes phenylalanine and bacterial therapeutics are being developed to lower phenylalanine levels in the GI-tract. The authors statement that ‘there remains a need for safe orally delivered therapeutic treatment options for PKU patients that will address the needs of the entire patient population regardless of age or genetic background’ is the correct assessment of the needs of PKU patients.

This is a very well written manuscript that overall describes a large number of complex experiments conducted clearly and concisely.

The following recommendations are made to improve the manuscript:

p. 20 line 370

‘This work focused on the application of an engineered aTF biosensor to improve the Phe-degrading enzyme PAL for a clinical PKU treatment strain.’

This sentence should be revised, the application of an engineered aTF biosensor is only one aspect of the large amount of work described.

- We thank the reviewer for this suggestion and agree that the sentence does not adequately describe of the scope of work covered in this manuscript. The text has been updated to “This work focused on the application of an engineered aTF biosensor to improve the Phe-degrading enzyme PAL for a clinical PKU treatment strain, followed by the development and preclinical testing of an improved live biotherapeutic, SYN1934.”

p.2 line 35 ‘A lead enzyme candidate from this screen was used in the construction of SYN1934, a chromosomally integrated strain containing the additional Phe-metabolizing and biosafety features found in SYN1618.’

AND p.17 line 306

‘SYN1934 was also engineered to express the same ancillary Phe degrading components of SYN1618, namely the high affinity Phe transporter, PheP, and the alternative Phe-degrading enzyme L-amino acid deaminase (L-AAD)¹⁰.’

This means that SYN1934 produces L-amino acid deaminase that catalyzes the conversion of Phe to phenylpyruvic acid and ammonia and phenylalanine ammonia lyase (PAL) which converts Phe to trans-cinnamic acid and ammonia.

The authors should discuss any potential toxicity from phenylpyruvic acid and ammonia and how they monitored for the potential toxicity and should mention what is known about trans-cinnamic acid toxicity.

- The reviewer certainly makes an important point that toxicity of these metabolites should be established clinically, though we feel that this information is not as relevant to the specific subject matter of the work presented here, so we are reticent to spend too much page space on this matter. However, we did add two references to support that trans-cinnamate, which is a naturally occurring metabolite, is non-toxic^{1,2}, as well as a reference supporting that phenylpyruvate is non-toxic³. As for Phenylpyruvate, this metabolite is normally elevated in PKU patients as Phenylalanine is directed to the alternative transaminase pathway; the amount of additional phenylpyruvate produced and then detected systemically by the strain is negligible since plasma levels were not significantly altered in treated PKU subjects. Additionally, we provide the reference demonstrating that SYN1618 was found to be safe and well tolerated in healthy subjects and in PKU patients. As for ammonia, this was measured in previous SYN1618 trials and we have not observed any increases in treated subjects.

p.10 line 203 ‘To compensate for instabilities introduced by mutations in and around the active site, evolutionary (position specific scoring matrix²⁹) and co-evolutionary (GREMLIN³⁰) analyses were performed to suggest favorable residues. As shown in Fig. 3a, the incorporation of phylogenetic analyses to introduce stabilizing mutations leads to designed mutations that are widely dispersed across StIA.’ (see methods section XX)

And corresponding method section on p.26 line 490 – 502

The phylogenetic analysis and the position specific scoring matrix (PSSM) are not sufficiently explained.

- We have added text to both the Results and Methods section describing the phylogenetic analysis (PSSM) in more detail.

p. 14 line 269 'In contrast to the very low activity observed when assayed at pH 5 ..., cells were able to recover from a 1 h exposure to pH 5 medium ... and demonstrated similar activity to control cells.'

The authors should describe how long the enzyme might be exposed to what pH while passing through the human GI system after oral administration.

- In parallel to our synthetic biology work on SYN1934, our lab has recently completed a comprehensive mechanistic modeling of SYN1618 to aid in prediction of strain activity in human subjects (see Figure 2 of Charbonneau, M., et. al., 2021⁴). We do plan on performing a similar mechanistic analysis for SYN1934 at some point in the future. We feel like the discussion around the complex and dynamic shifts of pH encountered during *in vivo* intestinal transit is a bit beyond the scope of this work, and a thorough explanation may detract from the work's focus on synthetic biology. However, the reviewer is correct that more information could be informative, so this recent aforementioned publication has been referenced in the discussion section to guide the reader to a location to obtain this information

Change to text: "Exposure to varying pH will undoubtedly be encountered by strains transiting the human GI tract. The effect this may have on strain activity is one parameter that has recently been mechanistically modeled for SYN1618⁴, and future work will aim to perform similar modeling for SYN1934."

p.16 line 291

'Of note, a higher K_M is consistent with a hypothesis throughout the project that a low K_M may correlate with more severe inhibition by the product TCA.'

This statement does not seem to be consistent with what a low K_M value means in enzymology.

- We thank the reviewer for this comment that highlights a likely source of confusion for readers. The hypothesis has to do with the structural similarity of Phe to the product TCA and demonstrated feedback inhibition with other PALs⁵⁻⁸. Since this hypothesis was not explored within the current work, it has been moved to the Discussion and the basis for the hypothesis has been further explained.

"In *in vitro* cell lysate experiments, mPAL showed significantly increased V_{max} as well as K_M . To mimic expression and activity in the final LBP as closely as possible, protein tags were not included in the construction of enzyme libraries. While the kinetic analysis on cell lysate does not allow for the decoupling of k_{cat} from enzyme concentration in the reported V_{max} values (see Table 1), normalization to total protein content is more representative of how a head-to-head comparison of therapeutic strains would be performed. Of note, a higher K_M is consistent with a hypothesis throughout the project that a low K_M (higher binding affinity of substrate to enzyme) may correlate with more severe inhibition by the product TCA, which is structurally very similar to the substrate Phe. Though the characterization of inhibition in wild-type StIA and mPAL was not undertaken in this work, feedback inhibition by TCA is a common trait in many other kinetically characterized PALs⁵⁻⁸ and is hypothesized for StIA based the increased activity observed in cell lysate compared to whole cells (Supplementary Fig. 1)."

p.19 'Fig. 5 In vitro and in vivo activity of SYN1934. Optimized PKU strain activity is compared to SYN1618. a, Schematic representation of SYN1934 chromosome shows key engineered genetic elements, including approximate location of genes encoding mPAL, PheP, LAAD, and deletion of dapA gene.'

Fig 5A rather looks like schematic representation of a bacterium than a bacterial chromosome
Is the schematic representation misleading or the figure legend incorrect?

- Replaced "chromosome" with "bacterium"

Reviewer #2 (Remarks to the Author):

Title: Improvement of a synthetic live bacterial therapeutic for phenylketonuria with biosensor-enabled enzyme engineering

Authors: Kristin J. Adolfsen, Isolde Callihan, Catherine E. Monahan, Per Jr. Greisen, James Spoonamore, Munira Momin, Lauren E. Fitch, Mary Joan Castillo, Lindong Weng, Lauren Renaud, Carl Weile, Jay H. Konieczka, Teodelinda Mirabella, Andres Abin-Fuentes, Adam G. Lawrence, Vincent M. Isabella

Recommendation:

Accept with minor revision

Overview and general recommendation:

Phenylketonuria (PKU) is a metabolic disease where patients have genetic defects in resulting in deficient phenylalanine (Phe) metabolism, which can result in neurological impairments if left untreated. The authors have previously developed a novel treatment for PKU in a form of engineered *E. coli* Nissle 1917 (EcN) strain (SYNB1618) expressing Phe metabolizing enzyme (phenylalanine ammonia lyase (PAL)), capable of turning Phe into a non-toxic product trans-cinnamate (TCA). In this work, the author has demonstrated a biosensor-based high-throughput screening method to screen and enrich a pool of engineered EcN with rationally engineered PAL enzymes. The plasmid-based biosensor was designed to detect TCA production from the EcN variants. In the absence of TCA, a sensor molecule bound the DNA operator, preventing the reporter *gfp* to be expressed. EcN variants were captured in microfluidic droplets allowing the biosensor system to detect whole cell PAL activity, followed by the use of fluorescence-activated cell sorter (FACS) to select cells with high GFP expression. After a few rounds of screens and enrichments, top hits were then characterized using two sets of experiments: 1) observing whole cell TCA production at various pH background to select variants that remained active with changing pH, and 2) exposing the cells to low pH and later observed the TCA production to select variants that were able to recover from a low pH

environment. The top two hits were then characterized for their Michaelis-Menten parameters via lysate kinetic experiment, and one final PAL hit was selected to generate EcN-derived SYN1934 strain. The strain was then compared with SYN1618 in simulated gastric model and in non-human primate studies. The authors found SYN1934's PAL activity to be about twice as effective as that of SYN1618 in these models.

Overall, this manuscript is well written, and the data provided are sufficient to support the flow of the story and the conclusions the authors made. The novel methods and findings would provide valuable insights into the future development of biosensor-based high-throughput screening platform of similar applications as well as the development of therapeutics on PKU using engineered EcN strains. The methodology presented in this work meets the standard practice in the field and is provided with enough details to be reproduced by others. Nevertheless, the following comments should be addressed during revision.

Major comments:

1. The authors reported most of the data in this paper as FOWT "fold over wild-type" which is great for showing method and strain improvements. However, the authors did not show the wild type data themselves along with their variances in most of these plots (exceptions will be discussed in major comment #2). This reduces the transparency in the data because we do not know how consistent the wild-type samples perform in these assays before they are taken the averages and used as benchmarks. I would recommend the authors to include the wild-type data points with correct error bars to all of the FOWT data plots (for the reason that I emphasized the word "correct" error bars, please see major comment #2 as well). I understand that the average values of the wild-type in these plots would be one because they are used as benchmarks for normalization. However, the error bars should still be reflective of the variance in the wild-type samples in each experiment.

- We thank the reviewer for this suggestion and agree that data transparency was lacking in the FOWT graphs of the original draft. This reviewer's comment has motivated us to repeat our analyses for many graphs and show wild-type instead of the early variant H133M I167K as a representative of assay reproducibility.

In the first draft, FOWT was calculated by:

- In graphs analyzed by "Plate-based activated biomass assay for TCA production from strains," normalizing by the average of wild-type replicates within the **same plate** (3 replicates in all cases except Fig. 1b, which had 2).
- In the pH assays of Fig. 4 and Supp. Fig. 4 and 5, where activated biomass was prepared in tubes, normalizing by the wild-type sample that was prepared on the **same day**. This is the source of the reviewer's major comment #2, since all three wild-type FOWT would be one using this method.

This previous method provided us with what we felt was the best analysis for differentiating improvements over wild-type, by isolating noise in improvement over wild-type from plate-to-plate or batch-to-batch effects prior to controlled bioreactor growth.

However, with the intention of publishing, we agree it is best to be transparent about these sources of noise. We have made the following updates:

- Fib. 1b top: Wild-type has not been added in this case, as it does not appear in the associated histogram. However, source data has been provided and is noted in the figure caption for data transparency.
- Fig. 1c: Converted FOWT calculations to normalizing by the average of all twelve wild-type controls across the four plates, rather than normalizing each plate by its respective 3 wild-type controls. Wild-type normalized in the same manner is now included on the plot to indicate noise across the experiment. H133M I167K, previously used as a benchmark for noise, has been removed upon the addition of wild-type. Additionally, source data has been provided.
- Fig. 4, Supplementary Fig. 4, and Supplementary Fig. 5: FOWT has been recalculated by normalizing by the average of the wild-type TCA production levels, rather than replicate-specific wild-type values, such that error bars are reflective of batch-to-batch variation. Figures have been updated, and source data is provided.
- Supplementary Fig. 2: Wild-type PAL has replaced H133M I167K as a representative of assay noise, and source data has been provided.
- We have added a description of FOWT normalizations to the methods. “For fold over wild-type (FOWT) normalizations, variant TCA levels were divided by the averaged wild-type TCA values from the same assay.”

One FOWT figure, Supplementary Fig. 3, has not been updated as it is the compilation of many experiments. We have added the following disclaimer to the figure caption:

“FOWT was calculated by normalizing by a wild-type control or controls in the same batch/plate. For much of the data in this plot, replicates were not performed, as we prioritized promising hits over thoroughly characterizing less improved variants.”

Similarly, Supplementary Table 1 is a compilation of many experiments, so we now refer the readers to accompanying Source Data for experiment-specific normalizations.

2. The authors did include wild-type samples in certain FOWT plot, namely supplementary figure 4e, f, g, h, and supplementary figure 5c. However, I noticed that all the error bars of the wild-type strain (EP2315) in these figures appear to be completely flat (or zero SD) which is not possible because I observe variances in the raw trace data in supplementary figure 4a, b, c, d and supplementary figure 5a, b. This could be an error during normalization process which left the wild-type data in the FOWT plots with just the average value with no variation between samples. Please restore the correct error bars to the wild type data in these figures to reflect the true variances of the wild-type strain.

- This concern has been addressed with the updates described in the last bullet point.

3. In supplementary figure 1, the authors showed whole cell and lysate TCA production data with increasing number of PAL copies within the cells, inferring the increasing amount of PAL proteins within the cells or lysate. However, the authors offered no evidence showing that increasing the copy number of PAL increased the amount of PAL protein production. Please

include the data showing the increase in PAL production as a function of copy number in a form of SDS-PAGE along with western blot or ELISA if possible.

- A new figure panel was added to supplementary figure 1 (Supplementary Figure 1b). An SDS PAGE gel was run using whole cells of wild type Nissle and the engineered PAL-expressing strains discussed in supplementary figure 1 (now Supplementary Figure 1a). Briefly, whole cells of each strain were normalized to an OD600 of 0.5 and boiled in LDS sample buffer for 15 minutes. Samples were run on a 4-12% Bis-Tris SDS PAGE stacking gel. A band corresponding to the size of PAL, absent in the wild type EcN lane, is clearly seen to increase in abundance upon increasing PAL expression/copy number, in line with the lysate data reported. This was repeated a second time and yielded identical results. Unfortunately we do not have an antibody capable of recognizing the PAL enzyme used in our strains, so we were unable to perform Western or ELISA analysis, but we feel that this gel conveys the same information that a Western would provide.

4. In line 291, the author wrote “Of note, a higher K_M is consistent with a hypothesis throughout the project that a low K_M may correlate with more severe inhibition by the product TCA.” This conveys that the authors theorized product inhibition in these enzymes but did not demonstrate the product inhibition behavior in the Michaelis-Menten plots. In figure 6, we do not observe the curve(s) turning downward due to product inhibition. I would suggest the authors re-run the Michaelis-Menten experiment with higher concentrations of Phe to induce product inhibition and reduction in the rates of reactions of these enzymes.

- We thank the reviewer for this suggestion. Unfortunately, the experiments as conducted are near the solubility limit of the L-Phenylalanine in water at room temperature, and we could only achieve approximately two- to three-fold higher concentration, depending on the behavior in the assay buffer. The product inhibition is theorized on the basis of being a common trait with other PALs⁵⁻⁸ as well as the increase in activity with increasing PAL expression in cell lysate but not whole cell context (Supplementary Figure 1).

Since this hypothesis was not explored within the current work, it has been moved to the Discussion.

“In *in vitro* cell lysate experiments, mPAL showed significantly increased V_{max} as well as K_M . To mimic expression and activity in the final LBP as closely as possible, protein tags were not included in the construction of enzyme libraries. While the kinetic analysis on cell lysate does not allow for the decoupling of k_{cat} from enzyme concentration in the reported V_{max} values (see Table 1), normalization to total protein content is more representative of how a head-to-head comparison of therapeutic strains would be performed. Of note, a higher K_M is consistent with a hypothesis throughout the project that a low K_M (higher binding affinity of substrate to enzyme) may correlate with more severe inhibition by the product TCA, which is structurally very similar to the substrate Phe. Though the characterization of inhibition in wild-type StIA and mPAL was not undertaken in this work, feedback inhibition by TCA is a common trait in many other

kinetically characterized PALs⁵⁻⁸ and is hypothesized for StIA based the increased activity observed in cell lysate compared to whole cells (Supplementary Fig. 1).”

5. Kinetic parameters determined from cell lysates will not be very accurate due to other protein contents in the cells. Besides potential inhibitors in the cell lysates, the authors cannot determine k_{cat} as they do not know the exact concentrations of the enzymes in these reactions. The V_{max} values are still dependent of the actual enzyme concentrations. I would recommend the authors to purify the proteins prior to the kinetic characterization. However, if these PAL proteins contain no tag for the purpose of being delivered as therapeutics, the kinetic parameters from the tagged versions of these enzymes (for the purpose of purification) would be acceptable and better than the current kinetic parameters from cell lysates.

- While we agree with the reviewer that an analysis on purified protein and determination of variant k_{cat} would be more thorough and of interest to readers, the authors feel that the undertaking is outside the scope of the present work given that SYN1934 has progressed into clinical trials. While we would certainly love the opportunity to purify these proteins for kinetic characterization, this work represents the culmination of a joint collaboration between Zymergen and Synlogic that has since met its conclusion, and the resourcing to perform this sort of characterization is not likely to be available for some time. Additionally, although it does not allow for the decoupling of V_{max} , untagged enzyme in cell lysate normalized for total protein content is more representative of the therapeutic whole cell context that patients will see. We have adjusted the Discussion to explicitly call out the limitations of the cell lysate approach and thank the reviewer for this comment.

“ To mimic expression and activity in the final LBP as closely as possible, protein tags were not included in the construction of enzyme libraries. While the kinetic analysis on cell lysate does not allow for the decoupling of k_{cat} from enzyme concentration in the reported V_{max} values (see Table 1), normalization to total protein content is more representative of how a head-to-head comparison of therapeutic strains would be performed.”

6. NHP study is missing a negative control where wild-type EcN is being fed to compare with SYN1618 and SYN1934. While missing this control does reduce the robustness of the data in figure 5c and 5d, it is understandable as it is difficult to have a large number of monkeys in a single experiment to explore all of the conditions. Moreover, the authors already ran this EcN control condition comparing to SYN1618 in Isabella V., et al (2018). I would suggest the author mention the negative control and how it underperformed SYN1618 and reference Isabella V., et al (2018) paper in the text.

- Changed original sentence from: “We have previously reported that plasma TCA can be used as a unique strain-specific biomarker to track PAL activity of strains⁹.”

To:

“We have previously reported that plasma TCA can be used as a unique strain-specific biomarker to track PAL activity of engineered strains; administration of wild-type EcN did not result in the detection of this metabolite⁹”

Minor comments:

1. In supplementary figure 1, please perform one-way ANOVA with multiple comparison (Ordinary ANOVA with Tukey test, assuming equal populations and variances) to convince the readers about the plateau of whole cell data and the increase of TCA production in lysate data with the increase of protein production.

- The data presented in this experiment were from an assay performed in biological duplicate, therefore we feel performance of ANOVA is unnecessary. However, the figure and legend have been modified to highlight this and to contain the individual data points.

The mechanical lysis employed to generate the lysate described in this figure was with a microfluidizer homogenizer, which allows for efficient lysis in the absence of detergents, but is also cumbersome and low throughput, which is the reason it was performed only in duplicate. However, we feel that these lysates provide the best picture of PAL activity stored in the cytosol, and their detergent/additive-free nature have the least chance of altering enzyme function. However, it is important to mention that although this particular set of data was from an assay performed in duplicate, our lab has performed 1000's of assays on PAL encompassing a wide range in expression level and we have consistently seen strain activity plateau at a similar rate that is apparently independent of expression.

2. The authors did not describe the details of experiments in supplementary figure 1 in the method. Detailed procedure of this specific whole cell and lysate assays should be described for the reader to comprehend the comparison between the two experiments. While the whole cell experiment was described in the method and the readers can assume a similar protocol was ran. There was no specific indication about the timepoint that this data set was taken. Moreover, there was no detailed explanation for this lysate assay in the method session. A similar lysate assay protocol was later described but that was for Michaelis-Menten characterization which included the protein normalization step, which I assume would be different in this experiment. Please describe these procedures in appropriate sub-sections in the method section.

- Apologies, the reviewer is correct that this information was not included in the text. This was an oversight. We added experimental detail to the methods sections as requested, under subheading “Comparison of whole-cells vs. lysates for examination of PAL expression, abundance, and activity”

3. Figure 1c, supplementary figure 2 and 3 lack the statistics that conveys the differences among the sorted groups. Please perform Welch ANOVA test with Dunnett T3 multiple comparison test, assuming different N in each group and wide range of variances, to determine significant differences among sorted populations and engineering rounds.

- The reviewer has highlighted important statistical analyses missing from the originally submitted draft. In performing the recommended tests, we found that the mock vs. top 1% and mock-mock vs. top 1%-top 1% comparisons of Fig. 1c were significant ($p < 0.0001$). In Supplementary Fig. 2, the pop 'n' sort methodology provides a statistically significant enrichment ($p = 0.0007$, mock-mock vs. top 1%-top 1%), whereas sorting directly from saturated liquid culture does not ($p = 0.1408$, mock-mock vs. top 1%-top 1%), more concretely highlighting the benefit of microfluidic encapsulation during incubation. The Results section of the main text as well as the "Pop 'n' sort methodology" section of the Supplementary Material now describe these findings.

For Supplementary Figure 3, we did not intend to imply any statistical differences across the different rounds of engineering, just to provide the interested reader with more background on the approaches taken throughout the course of the project, beyond the computationally designed combinatorial library that is the focus of the main text. Supplementary Fig. 3 shows the FOWT activity of unique sequenced PAL variants identified throughout the course of the project, and the accompanying text sheds light on the other enzyme engineering approaches used (e.g. random mutagenesis) and acknowledges progress made prior to the development of the sensor application strategy.

4. Line 48 of supplementary document mentioned "4 wild-type PAL colonies (green), and 4 H133M I167K colonies (red, an early hit with improved activity)". However, I'm only seeing the 4 H133M I167K colonies in green (not in red) on supplementary figure 2 and the wild-type PAL data is missing. I assume the authors reported FOWT and remove the wild-type data after restructuring the data. However, I would still recommend adding the wild-type data back as mentioned in major comment#1 and make sure the color in the caption matches with the actual color in the plot.

- We apologize for this error and thank the reviewer for catching it. Wild-type has replaced the early hit H133M I167K as an indicator of noise, and the caption has been corrected.

5. Line 193 – encapsulated cultures "became" saturated (missing a verb)

- We thank the reviewer for catching this error. The text has been updated.

6. Please consider rewrite this sentence in line 291 "Of note, a higher KM is consistent with a hypothesis throughout the project that a low KM may correlate with more severe inhibition by the product TCA." It's rather difficult to understand what the authors try to convey.

- We thank the reviewer for this comment that highlights a likely source of confusion for readers. The hypothesis has to do with the structural similarity of Phe to the product TCA and demonstrated feedback inhibition with other PALs⁵⁻⁸. Since this hypothesis was not explored within the current work, it has been moved to the Discussion and the basis for the hypothesis has been further explained.

"In *in vitro* cell lysate experiments, mPAL showed significantly increased V_{max} as well as K_M . To mimic expression and activity in the final LBP as closely as possible, protein tags

were not included in the construction of enzyme libraries. While the kinetic analysis on cell lysate does not allow for the decoupling of k_{cat} from enzyme concentration in the reported V_{max} values (see Table 1), normalization to total protein content is more representative of how a head-to-head comparison of therapeutic strains would be performed. Of note, a higher K_M is consistent with a hypothesis throughout the project that a low K_M (higher binding affinity of substrate to enzyme) may correlate with more severe inhibition by the product TCA, which is structurally very similar to the substrate Phe. Though the characterization of inhibition in wild-type StIA and mPAL was not undertaken in this work, feedback inhibition by TCA is a common trait in many other kinetically characterized PALs⁵⁻⁸ and is hypothesized for StIA based the increased activity observed in cell lysate compared to whole cells (Supplementary Fig. 1).”

7. It wasn't clear in the text why one out of the two final hit constructs was chosen over the other to be incorporated into SYN1934. Please describe the thought process along with relevant supporting information in the beginning of “Construction and testing of an improved live biotherapeutic strain, SYN1934” section.

- In actuality there was not a strong reason on why to use one variant over the other since they appeared to function identically and we didn't want to double the amount of work and characterization by building two strains. Of extremely minor concern was the addition of the Phe residue in the one variant that wasn't used (H133F). Incorporation of this variant over the other would ultimately increase the amount of Phe administered orally to PKU subjects, though there would only be concern if the cells lysed along transit. And even so, the amount of additional Phe from this variant is likely to be relatively trivial. However, the text was adjusted accordingly:

“Though there was no discernable difference between the 2 lead PAL variants characterized, the S92G_H133M_I167K_L432I_V470A variant, henceforth termed mPAL (for mutant PAL), was selected to construct SYN1934, an EcN-based live biotherapeutic (LBT) strain suitable for human dosing¹⁰. This variant was chosen over S92G_H133F_A433S_V470A because it did not incorporate an additional Phe that could ultimately contribute to dietary Phe load in dosed subjects if cells lysed during transit.”

This is really the only rationale worth mentioning in the text. But, as stated before, the amount of Phe is likely trivial, and we would leave it to the editor to choose the new language or the older language in the final version of the manuscript if ultimately accepted for publication.

8. More details should be provided about the in vitro GI simulation in the text part (Line 315-317). Upon reading the method, it seems this model mimics the gastric. Is it possible to simulate other parts of the gut besides the stomach? I assume the pH and other environment factors would be drastically different from stomach to duodenum, jejunum, and ileum. It would be great to see the activities of the three strains at different parts of simulated gut environments to see the location where the cells would be most active.

- The reviewer is correct that this is a simulated gastric model, however it is important to also mention that the stomach is buffered to near neutral pH in this model, similar to the way that our strain dosing formulation buffers the human stomach (or NHP stomach in the case of our preclinical studies). The other upper GI compartments would be unlikely to have a pH significantly different from that of a buffered gastric environment. We do feel that the gastric simulation provided in this manuscript is an accurate representation of the strain activity expected in vivo, as the increase in TCA production noted in SYN1934 in the gut simulation accurately predicted the increase in biomarker recovery observed in vivo in non-human primate studies. The reference to Charbonneau, M., et. al., 2021⁴ is also provided in the revised version of this manuscript to guide the reader to a more comprehensive overview of our gut simulation models.

9. In figure 5b, please perform one-way ANOVA with multiple comparison (Tukey method) to demonstrate the differences between the three strains in simulated GI model

- PRISM was used to perform this analysis on the data set and the following language was used in the text to address Figure 5B:

“...resuspended SYN1934 demonstrated a significant increase in the rate of TCA production of approximately 2-fold compared to SYN1618 (Fig. 5b; one-way ANOVA followed by Tukey’s multiple comparison test, $p < 0.001$)”

10. Line 653: the method did not mention the addition of arabinose for induction of LAAD in SYN1934 strain

- This was an error that we thank the reviewer for pointing out. Both SYN1618 and SYN1934 were induced for LAAD activity to simulate the production and lyophilization of drug product, even though LAAD activity was not measured in these assays due to the lack of any robust in vivo biomarker to determine the extent of its function in vivo. The text has been modified to include the LAAD induction of the process.

11. Line 713: missing a negative sign at “-80”

- We thank the reviewer for catching this error. The text has been updated.

12. Line 764: “The complete genome sequence of SYN1934 is available under BioProject ID: XXXXXXX” Please fill out the correct ID.

- The whole genome sequence of SYN1934 has been submitted to NCBI and has received BioProject ID# PRJNA749270. This has been added to the text. The sequence is set to be published if or when the work presented in this manuscript is published. The genome has been given the accession number CP080246.

13. Line 707: the text reads “resuspended and administered PO 1011 Live Cell doses to each animal”. Please describe PO abbreviation or change the word to “orally”

- Deleted “PO” and replaced with “orally”

1. Hoskins, J. A. The occurrence, metabolism and toxicity of cinnamic acid and related

- compounds. *J. Appl. Toxicol.* **4**, 283–292 (1984).
2. Hoskins, J. A. & Gray, J. Phenylalanine ammonia lyase in the management of phenylketonuria: the relationship between ingested cinnamate and urinary hippurate in humans. *Res. Commun. Chem. Pathol. Pharmacol.* **35**, 275–282 (1982).
 3. van Spronsen, F. J. *et al.* Phenylketonuria. *Nat. Rev. Dis. Prim.* **7**, 1–19 (2021).
 4. Charbonneau, M. R. *et al.* Development of a mechanistic model to predict synthetic biotic activity in healthy volunteers and patients with phenylketonuria. *Commun. Biol.* **4**, 1–12 (2021).
 5. Sato, T., Kiuchi, F. & Sankawa, U. Inhibition of phenylalanine ammonia-lyase by cinnamic acid derivatives and related compounds. *Phytochemistry* **21**, 845–850 (1982).
 6. Appert, C., Logemann, E., Hahlbrock, K., Schmid, J. & Amrhein, N. Structural and catalytic properties of the four phenylalanine ammonia-lyase isoenzymes from parsley (*Petroselinum crispum* Nym.). *Eur. J. Biochem.* **225**, 491–499 (1994).
 7. Camm, E. L. & Towers, G. H. N. Phenylalanine ammonia lyase. *Phytochemistry* **12**, 961–973 (1973).
 8. Pridham, J. B. & Woodhead, S. Multimolecular forms of phenylalanine-ammonia lyase in *Alternaria*. *Biochem. Soc. Trans.* **2**, 1070–1072 (1974).
 9. Isabella, V. M. *et al.* Development of a synthetic live bacterial therapeutic for the human metabolic disease phenylketonuria. *Nat. Biotechnol.* **36**, 857–864 (2018).
 10. Charbonneau, M. R., Isabella, V. M., Li, N. & Kurtz, C. B. Developing a new class of engineered live bacterial therapeutics to treat human diseases. *Nat. Commun.* **11**, 1–11 (2020).

Reviewers' Comments:

Reviewer #2:

Remarks to the Author:

The authors have addressed all my comments during the first review. As mentioned previously, this is a very well-written manuscript that contains a novel therapeutic idea and a great depth of data analysis. This work could have a strong impact on the future of PKU treatment and I recommend this article for publication at Nature Communications.

Response to Reviewers:

REVIEWERS' COMMENTS

Reviewer #2 (Remarks to the Author):

The authors have addressed all my comments during the first review. As mentioned previously, this is a very well-written manuscript that contains a novel therapeutic idea and a great depth of data analysis. This work could have a strong impact on the future of PKU treatment and I recommend this article for publication at Nature Communications.

We thank the reviewer for their kind words and endorsement of the work